# Nanoparticles and Mesenchymal Stem Cell (MSC) Therapy for Cancer Treatment: Focus on Nanocarriers and a si-RNA CXCR4 Chemokine Blocker as Strategies for Tumor Eradication In Vitro and In Vivo

**DOI:** 10.3390/mi14112068

**Published:** 2023-11-07

**Authors:** José Joaquín Merino, María Eugenia Cabaña-Muñoz

**Affiliations:** 1Departamento de Farmacología, Farmacognosia y Botánica, Facultad de Farmacia, Universidad Complutense de Madrid (U.C.M.), 28040 Madrid, Spain; 2CIROM—Centro de Rehabilitación Oral Multidisciplinaria, 30001 Murcia, Spain; mecjj@clinicacirom.com

**Keywords:** nanoparticles, mesenchymal stem cells (MSC), gene therapy, homing, nanocarriers, exosomes, PLAG, glycolic acid, CXCR4, chemokine blockers (AMD-3100, plerixafor), tumors, stem cell therapy: CXCL12 (SDF-1 alpha), nanocarriers

## Abstract

Mesenchymal stem cells (MSCs) have a high tropism for the hypoxic microenvironment of tumors. The combination of nanoparticles in MSCs decreases tumor growth in vitro as well as in rodent models of cancers in vivo. Covalent conjugation of nanoparticles with the surface of MSCs can significantly increase the drug load delivery in tumor sites. Nanoparticle-based anti-angiogenic systems (gold, silica and silicates, diamond, silver, and copper) prevented tumor growth in vitro. For example, glycolic acid polyconjugates enhance nanoparticle drug delivery and have been reported in human MSCs. Labeling with fluorescent particles (coumarin-6 dye) identified tumor cells using fluorescence emission in tissues; the conjugation of different types of nanoparticles in MSCs ensured success and feasibility by tracking the migration and its intratumor detection using non-invasive imaging techniques. However, the biosafety and efficacy; long-term stability of nanoparticles, and the capacity for drug release must be improved for clinical implementation. In fact, MSCs are vehicles for drug delivery with nanoparticles and also show low toxicity but inefficient accumulation in tumor sites by clearance of reticuloendothelial organs. To solve these problems, the internalization or conjugation of drug-loaded nanoparticles should be improved in MSCs. Finally, CXCR4 may prove to be a promising target for immunotherapy and cancer treatment since the delivery of siRNA to knock down this alpha chemokine receptor or CXCR4 antagonism has been shown to disrupt tumor–stromal interactions.

## 1. Homing: Recruitment of Mesenchymal Stem Cells (MSCs) toward Tumor Site and Damaged Tissues

Mesenchymal stem cells (MSCs) encompass a group of multipotent progenitor stromal cells that are privileged from an immune point of view. The homing of stem cells toward damaged tissues (“guiding or directing of said SCs”) toward tumor sites is induced by chemotactic gradients by chemokines, which promote proliferation of stromal cells and induce immunomodulatory/angiogenic effects by paracrine released factors. The homing of MSCs toward injured areas enhance repair capacity of stromal cells under amatorytory conditions [1]). The stromal-derived factor SDF-1α (also termed CXLC12) binds to CXCR4 chemokine receptor and facilitates stem cells (particularly MSCs) homing toward damaged tissues [2]. SDF 1 alpha is a pleiotropic chemokine that induces apoptosis and tumor growth and also recruits neural stem cells toward damaged tissues [3]. Indeed, this SDF1 alpha ligand is released by tumor cells or damages cells (in particular, recruits MSC cells, see Figure 1) [4]. MCP-1 (macrophage colony protein-1) is another released chemokine, in conjunction with more factors such as hypoxia-inducible factor (HIF-1 alpha); integrin alpha4-beta-1 and molecules such as CD44, PG-E2 and PPAR-gamma (peroxisome proliferator receptor activated factor) also contribute to the homing of MSCs [1,5,6,7].

In addition, MSCs’ homing may be influenced by biomechanical factors [8,9,10]. Therefore, exogenous factors that favor the homing of MSCs can facilitate the integration of transplanted MSC cells, generating a more functional graft within the damaged tissues [11,12]. However, eventual immune recognition of allogeneic MSCs could limit their clinical efficacy due to a possible immune rejection [13]; for example, MSC transplant promoted anti-inflammatory effects via TSG-6 by reducing neutrophil migration in a rodent model of cerebral ischemia [14,15].

How does the homing occur?

The homing process (especially of MSCs) involves a series of steps:(1)Initial anchorage by selectins (rolling).(2)Adhesion and activation by chemokines like CXCR4 and/or cell adhesion molecules (e.g., VCAM-1: vascular adhesion molecule-1).(3)Retention by integrins.(4)Diapedesis or transmigration.(5)Extravascular migration by CXCR4/SDF1 alpha-dependent levels (chemotactic gradient), which directs MSC toward the tumor site.(6)Stromal or mesenchymal MSCs release CXCR4/SDF-1 chemokines (see Figure 1).

On the other hand, the transfection of CXCR4 on MSCs enhances their homing capacity in vitro [12,16,17]. In the niche of stem cells, the secretion of trophic factors and chemokines also promote proliferation of stromal cells, and remodelates the extracelullar matrix [18,19], see Figure 2; The proximity of the SC niche to blood vessels also induce vascular factor release (VEGF, angiopoietin) involved in tissue repair. On the other hand, MSC-derived exosomes are essential for communication between cells are nanovesicles derived from membrane (a cholesterol-rich lipid bilayer) [7,20], which release certain miRNAs-iR-23a, miR-26b, miR-125b, miR-130b, miR-140, miR-203a, miR-223, miR-224, miR-320a; tetraspanins (CD68 and CD8) are markers of exosomes [21]. 

The balance between pro-inflammatory M1-released signals by macrophages and the control of M2 anti-inflammatory responses regulate the immunomodulatory effects of MSC. For example, macrophages M1 are scavengers able to remove cellular debris, and neutrophils act as early invaders of the niche by releasing proinflammatory cytokines. Indeed, MSCs can inhibit apoptosis and orchestrate immunomodulatory effects in these neutrophils [22].

On the other hand, the use of pharmacological antagonists such as AMD3100 or plerixafor (a human CXCR4 antagonist) blocks the CXCR4 chemokine receptor. In normal conditions, hematopoietic stem cells (HSC: CD 34 positive cells) are retained in the bone marrow when SDF1 Alpha binds to CXCR4 on stromal cells; Thus, plerixafor as CXCR4 human antagonist enhanced hematopoietic stem cell mobilization (CD-34 positive cells) from the bone marrow into the circulation (bloodstream). This rapid increase in neutrophils into circulation via plerixa for treatment has been demonstrated in a clinical phase trial. In addition, the G-CSF (Granulocyte Colony Stimulating Factor) also mobilizes HSC cells into the bloodstream [23], see Figure 3.

(A) The interaction of SDF1 Alpha = CXCL12 with its CXCR4 chemokine receptor retains hematopoietic stem cells (HSC-CD34+ cells) within the bone marrow; however, (B) when CXCR4 is blockade by plexarifor, HSC mobilization is increased from the bone marrow into the bloodstream [23]. 

## 2. Mesenchymal Stem Cells as Drug Vehicles: Nanoparticles as Drug Carriers

Nanoparticle-based biomimetic systems provide an alternative treatment for drug delivery, but they still lack active targeting capability. Nanoparticle technology has managed to improve the capacity of MSCs as drug carriers by enhancing their homing as antitumor agents [24]. The progression in the field of genetic engineering, biomaterials [25], and nanotechnology has contributed to a better understanding on the role of mesenchymal (MSC), as drug carriers with nanoparticles [26]. In particular, human mesenchymal stem cell (hMSC)-based therapy can enhance the homing of exogenous MSCs, which could be drug carriers [27]; this enhanced capacity of MSC could be interesting from a clinical viewpoint [12]; cancerous stem cells have the ability to self-renew and contribute to tumor metastasis and pharmacological resistance to drugs could be solved by MSC treatment against cancer since MSCs could release cytotoxic agents in the tumor microenvironment for its eradication [28]; In fact, MSCs could act as a “Trojan horse” able to transport the oncolytic viral load to disseminate tumor beds for its eradication. Another strategy is the use of cell membrane-coated nanoparticles [29]. In this way, even MSC-derived producers such as exosomes from bone marrow (MSC-BMSC) could prevent osteosarcomas growth in vitro [16]. The CXCR4 inclusion in these nanoparticles could be an effective strategy for enhancing the homing of MSC toward tumors such as we mention earlier [30]; the ability of MSCs as drug carriers is also enhanced by spheroidal nanoparticle technology, which facilitates the transport of “drugs/therapeutic agents” within MSCs [31]. However, a limited capacity of homing towards the target tissue makes its clinical translation by reducing the release of “loaded drug” inside the tumor site. In addition, the risk of low viability of MSCs decreases the capacity to transport drugs [32]. The monitorization of transplanted MSC cells in vivo with nanoparticles is possible; however, these NP can provoke cell aggregation, infections, or induce differentiation to an unexpected cell type or event risk of embolism in case of i.v infusion. These limitations reduce the efficacy of NP and consequently decrease the antitumoral capacity of MSC [33]. 

CXCR4 plays a role in chemotaxis, adhesion, and migration, which contribute to tumor metastasis by a CXCR4 overexpression signaling pathways in tumor cells; these NP are carriers of si-RNA for CXCR4 while the CXCR4 blockade by AMD-3100 chemokine blocker can prevent tumor growth in vitro models of cancer [34]; in fact, genetically engineered mesenchymal stem cell (BM-MSC: bone marrow mesenchymal stem cells) membrane-derived nanoparticles that overexpress CXCR4, enhanced the migration capacity toward the tumor locations; these cancer cells are identified using fluorescent NP in the site of tumor. From this perspective, several studies have demonstrated antitumoral effects of MSC although few papers reported its conversion to tumoral cells. This review describes the role of si-RNA for NP CXCR4-MSC as an antitumoral drug.

### Does Transplanted Mesenchymal Stem Cells (MSC) Exert Antitumoral or Favor Tumorigenesis?

MSC releases several paracrine factors involved in proliferation or repair (i.e., bFGF fibroblast growth factor), Platelet growth factor (PDGF), neurotrophic factor (NTF), GDNF (glial neurotrophic factor), BDNF: brain-derived neurotrophic factor), endothelial growth factor (VEGF), HGF (hepatocyte growth factor), Insulin Growth factor (IGF-1) [7,35]. For example, secreted factors by fat-derived mesenchymal stromal cells (MSC-ADSC) protect chondrocytes in rodents of osteoarthritis by increasing HFG, and these MSCs also promoted immunomodulatory effects [36] and converted pro-inflammatory M1 monocytes into an anti-inflammatory M2 state [37]; Figure 4 summarizes the effect/s of MSC on cell adhesion, migration and proliferation markers.

The immunoregulatory activity of MSCs is mediated through secreted cytokines by antigen-presenting cells (IFN γ, IL-1α, IL-1β, and TNF-α) [38] and several factors (IDO, TSG6, NO, IL-10, CCL2, galectins, PGE2 or TGF-β) [39,40,41]. The activation of MSCs by Tumor necrosis factor α (TNF-α), IL-1β, nitric oxide (NO), liver growth factor (HGF), and VEGF promoted immunomodulatory effects in MSC [42], see Figure 4. 

The few studies that detected protumor effects of MSC, in general, are explained by the transformation of these MSCs into cancer cells [43,44]. Several factors like SDF-1 alpha (as occurs in gynecological tumors), HDGF (hepatoma-derived growth factor), VEGF, bFGF, or MCP-1 (monocyte chemotactic protein-1) are also involved in the proliferation of different cell types. These released factors by MSCs enhances the migration toward the tumor site in a CXCR4/SDF-1 dependent manner [45]; in addition, the angiogenesis by VEGF increases the proliferation of tumor cells by forming new blood vessels. In this case, MSCs can enhance tumor proliferation, inducing metastasis by the conversion of MSCs into tumoral cells [46,47]. For example, bone marrow-derived MSCs can progress to gastrointestinal epithelial cancer by *Helicobacter hepaticus* infection in mice [43,44]. 

MSCs exert antitumoral effects by regulating cell cycle proteins, reducing angiogenesis, promoting apoptosis of cancerous cells as well as immunomodulatory effects [48,49,50,51,52,53,54,55]. In fact, when MSCs are co-transplanted in a glioblastoma rodent model of cancer, both angiogenesis and tumor progression are abolished in different types of cancers (e.g., from colorectal to hematological type) [56].

To date, clinical studies have reported the safety of MSCs without tumor formation in patients. In a meta-analysis of 36 studies, including eight randomized trials with adequate control, the absence of tumors has been demonstrated after MSC transplantation (n = 1012 patients [57]. In this way, the immunological status of MSC-transplanted animals with NP with different nanocarriers becomes relevant [58]. For example, bone marrow-derived MSCs (BM-MSCs) can spontaneously transform into malignant fibrosarcoma after systemic administration to immunocompromised mice. However, the long-term MSC transplantation did not show tumors in tissues [58]. Fibrosarcoma formation was detected after transplanting MSCs in immunocompetent mice with spontaneous p53 mutations [59]. Two further studies conducted with stem cells observed malignant transformations of MSC cells in *Cynomolgus macaques* [60,61]. These chromosomal aberrations seem to be associated with cellular senescence [62,63]. In any case, the possibility of rare tumorigenic transformations should not be ignored and cytogenic aberrations should be tested in cases of long-term MSC transplantation [64]. 

## 3. MSC as Vehicles for Drug Delivery against Tumoral Cells

The International Organization for Standardization (ISO) textually defines a nanomaterial as a “material with any external dimension in the nanoscale or having an internal structure or surface structure in the nanoscale (1–100 nm)”. The European Commission adopted a definition for a nanomaterial (2011) as follows: “A natural, incidental or manufactured material containing particles, in an unbound state or as an aggregate or as an agglomerate and where, for 50% or more of the particles in the number size distribution, one or more external dimensions is in size range 1–100 nm”. The United States Food and Drug administration (US FDA) states that nanomaterials are “materials up to one micron if these ones exhibit properties or phenomena that are attributable to its dimensions”. In this review, a size range between 1–1000 nm is considered for nanomaterials and nanoparticles (NPs) [65,66]. Metallic nanoparticles, with a dimension of 1–100 nm, were first described by Faraday M (1987) in an aqueous solution when he recognized the formation of ruby colored-AuNPs after a reaction of gold salt [65]. NPs can be used in several biomedical applications because of their unique physicochemical properties (high ratio of surface area to volume, surface plasmon resonance (SPR), presence of edges and corners, and electron storage capacity, among others [67,68]. NPs have several sized and shapes and can be prepared with different chemical, physical, and biological methods [69]. For example, gold (Au), copper (Cu), Silver (Ag) are pure metals metallic NPs, while cerium oxide NPs (nanoceria), Silicate NPs (SiNPs), zinc oxide (ZnO), titanium oxide (TiO_2_), and iron oxide NPs (SPIONs) are metal oxide. Accordingly, scientists have attempted to use stabilizing agents, such as polyacrylic acid, polyvinylpyrrolidone, and polyvinyl alcohol, which are adsorbed onto the surface of particles; as a consequence, the minimized particle aggregation takes place and form a stable solution of the metallic NPs [70]. The tiny size of nanomaterials allows efficient interaction with the cell surface of cell components, including stem cells [71]. Drug delivery, diagnostics, imaging, cancer therapy, anti-microbial, and antiangiogenic are several biomedical applications of NP that would allow better therapeutic use of metallic NPs in the field of regenerative medicine. The Impact of metallic nanoparticles on stem cell proliferation and differentiation has been revised under different pathological conditions [72]. Collectively, particles with tailored physicochemical properties can exert differential influence on stem cell differentiation and signaling pathways, which regulate stem cell differentiation and proliferation by regulating inflammation and oxidative stress [72,73]. As an example, silica-coated magnetic nanoparticles (NP) decreased the migratory capacity of human bone marrow-derived mesenchymal stem cell (hBM-MSC) by reducing membrane fluidity and also affects levels of proteins involved in focal adhesion [74]. In addition, biophysical properties can alter nanomaterials effects on stem cells and reduce the membrane fluidity or reduce 10%or more the viability of hBM-MSCs labeled with silica-coated magnetic NPs (stained with rhodamine B isothiocyanate); MNPs-SiO increased the oxidative stress in hBM-MSCs as compare to controls. The major mechanism underlying nanoparticle-induced cell shrinkage and abnormal formation of focal adhesions involved cytoskeletal depolymerization [73]. However, in this study, MNPsSiO_2_ (RITC)-induced lipid oxidation in a concentration-dependent manner without specific interaction with cytoskeletal proteins. Thus, the migratory capacity of hBM-MSCs is impaired by MNPs-SiO_2_ (RITC) [73].

It is known that passive transport and endocytosis are two key processes required for the reuptake of NP by cells in general, including MSC [65]. The NP’s physicochemical properties (e.g., size, shape, and stiffness and surface charge), biocompatibility and bioaccumulation, colloidal stability, degradation rate, e.g., solubility as well as the route of administration (e.g., intravenous, oral, intranasal, dermal) are cue factors neccesary for a final good penetrance within the tumor location [74]. These factors can influence the cell–NP interactions and subsequent uptake. The principal parameters to take into consideration for NP physicochemical properties on endocytic mechanisms are revised by several authors [73,74,75].

The interaction between different NPs and cells depends not only on the shape of the NPs but also on the cell type. The particle size and shape influence the biological function and the NP internalization into the cells; Additionally, other factors such as stiffness, the hydrophilic or hydrophobic properties are important for NP-stem cell interactions [71]. In fact, the uptake of NPs is inversely correlated with the particle sizes; in fact, the higher uptake of smaller size NPs (30–50 nm) was reported, compared to bigger size NPs (50–200 nm) with less cellular internalization [76,77]. The interaction between different NPs and cells depends on the shape of the NPs and the type of cell. Shape can increase or decrease uptake of NPs because the sphero-cylinders are more efficiently endocytosed compared to spheres of the same diameter. The membrane curvature, less available receptor binding sites, and surfactant molecules could also explain the lack of multivalent binding to the receptors [78,79]. In this way, spherical-shaped NPs showed higher uptake rate than non-spherical NPs [79,80]. The chemical modification of NPs by increasing the softness and the hydrophobicity increases the rate of internalization [81]. In addition, a particle surface charge has been associated with differential cellular internalization since a positive charge can quickly enter the nucleus and avoid the lysosomal degradation of NP; conversely, those particles with neutral or negative charges access into the lysosomes instead of at the perinuclear region [82,83]. As an example, the anti-microbial activity of TiO_2_ NPs was elevated when combined with gold Au/TiO_2_ nanocomposite by alteration in the surface charge of TiO_2_ NPs when were conjugated with gold [84]. Finally, the potential interference of food supplements could affect the NP uptake capacity [74,85]. As an example, we highlight the endocytosis and cytotoxicity of zinc oxide (ZnO) NPs and vitamin C co-exposure affect the uptake of NP by neural stem cells [86]. 

The ability to carry drugs is enhanced when MSC cells are exposed to nanoparticles (NP) [8]. Several studies demonstrated that certain drugs could be internalized by caveolins or clathrin receptors in MSC (present on the cell membrane); caveolins are essential proteins that form invaginations of the cell membrane and also play a role in intracellular transport. Drugs can also be internalized by pinocytosis or macropinocytosis [87]. 

On the other hand, intracellular degradation of NPs often takes place in the lysosomes following the endocytosis pathway. The degradation process is affected by the acid pH in the endolysosomes (close to pH 4.5); these lysosomes strong proteolytic activity regulates redox signaling mechanisms by glutathione and metallothioneins; thus, this environment favors the dissolution of several metallic NPs (iron oxides, ZnO, CuO, Ag, and AuNP); The pace of NP degradation within the cells showed similarities between the cell responses to metallic NPs and ionic forms [74]. The Aurosome formation illustrates that ions and NPs of the same metal seems to been processed by cells in the same manner. For example, Au ions and NPs have a common intracellular fate in aurosomes, and their internalization and location in lysosomes appear quickly after exposure to ionic Au but within days to months after exposure to Au NPs [74]. 

The modification of the particle surface by the addition of functional group can modulate its cellular uptake and could induce toxicity in the cells by TiO_2_ nanorods functionalized with various functional groups, such as carboxyl groups (–COOH), poly (ethylene glycol) (–PEG), and amines (–NH_2_), which showed a variation in their uptake by rat bone marrow-derived MSCs (rBM-MSCs) [75,88,89]. In fact, TiO_2_–NH_2_ nanorods and the core nanorods showed the highest rate of cellular internalization compared with TiO_2_–COOH and TiO_2_–PEG nanorods with lower uptake. However, the TiO_2_ core nanorods are toxic and increase reactive oxygen production [85,89]. Another study has demonstrated that delivery of methotrexate, (anti-cancer drug), was conducted by binding on the carboxyl group with the surface of AuNP [89]. On the other hand, the anti-microbial activity of TiO_2_ NPs increased when they were combined with gold (Au/TiO_2_ nanocomposite) [84]. Thus, the application of metallic NPs allows drug delivery for cancer therapy [86,90].

To prepare efficient magnetic nucleic acid carriers, it is necessary to improve the endocytosis efficiency of PEGylated magnetic nanoparticles. For example, heptafluorobutyryl-polyethylene glycol-polyethyleneimine (FPP) are used to coat magnetic nanoparticles (MNPs) to obtain magnetic nanocarriers FPP-MNPs. The fluorinated cationic polymer-coated magnetic nanoparticles FPP-MNPs can be loaded with siRNA [91,92]. The NP used in the core can be designed for future applications (e.g., anti-cancer drug delivery). These membrane-coated NPs have been shown to preferentially accumulate at tumor sites [4]. Via these mechanisms, MSCs could carry antitumor drugs to localized sites of the tumor [5]. In addition, the use of natural polymers, which can be “packaged” on the surface of MSCs, enhances their drug delivery [71]. For example, supermagnetic iron oxide nanoparticles (SPIONs: a type of iron oxide NPs: SPIONs) possess superparamagnetism property when are subjected to an external magnetic field, see Figure 5) [72,93,94,95]. SPIONs nanoparticles are considered as being a safe nanomaterial for stem cell labelling. For example, coating SPIONs Superparamagnetic iron oxide NPs with dextran (DEX) and then labeling stem cells promoted myogenic differentiation under a pulsed electromagnetic field (MyoG and Myh2 marker) [96]. However, the special physicochemical properties of SPIONs induced cytotoxicity by affecting cellular components (DNA, nucleus, and mitochondria) [97]. The surface chemistry alteration of SPIONs by citrate significantly hampered osteogenic differentiation of MSCs and also decreased the expression of induced osteogenic differentiation-related genes [89]. Conversely, high concentration of SPIONs (Ferucarbotran: 300 µg/mL) abrogated the osteogenic differentiation and enhanced the cell migration of MSC, which was mediated via the activation of β-catenin, and also matrix metalloproteinase 2 (MMP2) signaling pathways. Therefore, free iron is implicated in SPION-induced inhibition of the osteogenic differentiation of hMSCs [98]. Conversely, SPIONs (Feridex)-labeled hBM-MSCs did not affect the osteogenic or adipogenic differentiations, although they suppressed the chondrogenic differentiation [99]. 

On the other hand, external stimuli such as magnetic fields can influence the high rate of particle internalization into stem cells [94]. In this way, the magnetic field-induced assembly (stripe-like) of magnetic SPIONs was exploited for the conversion of primary mouse bone marrow cells into osteoblasts [100]. The interface between SPION magnetic assemblies and the cells promote osteogenic differentiation rather than the particles internalization into the human bone marrow stem cells (BM-MSCs) [101]. In addition, SPIONs induced hMSCs proliferation by regulating cell cycle-related proteins under oxidative stress [102]. As an example, SPION-induced gap junction communication between cardiomyoblasts while MSCs protected against myocardium infarct in mice [103]. 

Rafieepour et al. reported toxicological effects of Fe_3_O_4_ and SiO_2_ single and combined exposure of magnetite (Fe_3_O_4_) in cells were exposed to different NP concentrations (10, 50, 100, and 250 mg/mL), which were simultaneously for 24 h and 72 h. In contrast, the effect of combined exposure to Fe_3_O_4_ and SiO_2_ NPs is less toxic by the accumulation of intracellular proteins, forming a protein corona. In addition, Fe_3_O_4_ could induce the synthesis of cellular proteins, forming a protein corona on the surface of silica NPs and thereby reducing its cytotoxicity [104]. Thus, the formation of a corona can reduce cellular uptake of functionalized NPs by shielding the ligands from binding to their receptors, which prevent the recognition by transferrin receptors. Consequently, the composition of a protein corona is an important determinant of NPs fate and their cellular internalization [105]. However, the corona composition varies depending on the nature of the biological fluids in which NPs are dispersed [106]. The preparation of engineered NPs with the desired functional group is a tool for biomedical applications [107,108]. The internalization rate of polystyrene NPs and polystyrene NPs functionalized with an amine group in MSCs was studied in vitro [109]. Amino-functionalized polystyrene NPs showed faster internalization and higher cellular uptake in comparison to unfunctionalized polystyrene NPs. For example, TiO_2_ NPs are toxic for MSCs in a size-dependent manner by reducing cell migration, promoting a lack of cell membrane integrity, and suppressing osteogenic differentiation [110]. TiO_2_ nanotubes larger than 50 nm showed a drastically decrease in the proliferation and differentiation of MSCs [111]. In addition, TiO_2_–COOH nanorods abolished the osteogenic differentiation of rat BM-MSCs [89]. 

Collectively, particles with tailored physicochemical properties can exert differential influence on stem cell differentiation and affect its proliferation. Interestingly, the stem cell differentiation capacity is influenced by ROS production [72,111]. 

On the other hand, MSCs can interact with two different classes of nanoparticles (NP), either those containing polylactic acid or lipid nanocapsules. In addition, covalent, hydrophobic, or electrostatic interactions on the surface of MSCs enhance the drug-carrying capacity [112,113]. For example, glycolic acid polyconjugates enhance nanoparticle drug delivery in human MSCs [114]. In addition, covalent, hydrophobic, or electrostatic interactions on the surface of MSCs also increase their drug-carrying capacity [114]. For example, glycolic acid polyconjugates enhance nanoparticle drug delivery in human MSCs [114] while coating PLGA-type nanoparticles with the macrophage membrane promotes anti-angiogenesis effects [115]. PLGA nanoparticles can be loaded with saikosaponin D, a compound with potential therapeutic properties against cancer; the use of red blood cells as biomimetic membranes guarantee a prolonged blood circulation time of the NPs. In addition, enzymatically active methods as the use of catalase on the erythrocyte membrane [116] could metabolize endogenous hydrogen peroxide from the tumor as a strategy for tumor eradication [117]; thus, the cell membrane-engineered NPs effectively inhibited tumor growth (e.g., breast cancer metastasis in vitro and in vivo) by blocking angiogenesis (a process involved in tumor dissemination). In turn, labeling with fluorescent particles (such as coumarin-6 dye) allows the identification of tumor cells in tissues by fluorescence emission [118]. Another study reported that carbon nanoparticles can reduce the viability of human MSC cells [73,119].

From this perspective, nanotechnology facilitates the migration and distribution of MSCs within tissue with high sensitivity and resolution [119,120,121,122,123,124,125]. Paclitaxel-loaded polymeric NPs (anticancer drug) as drug delivery system significantly inhibits tumor growth and also suppresses lung metastasis. Platelet membranes as nanocarriers can be co-loaded with tungsten oxide nanoparticles (W18O49) and metformin (PM-W18O49-Met NP) to treat lymphomas; in fact, PM-W18O49-Met significantly inhibited tumor growth by inducing apoptosis of lymphoma tumors in vitro and in vivo. In this study, metformin reduced tumor oxygen consumption and enhanced the therapeutic effects of W18O49 [121]. For example, gold nanoparticles can induce their differentiation into osteocytes, a property attributable to such gold nanoparticles [125]. Gold nanoparticles loaded with the IR775c label facilitate the monitoring and distribution of MSC cells ¨in vivo¨ by fluorescent particles (such as coumarin-6 dye) in tumor cells with high sensitivity and resolution [118,120,121,126]. Interestingly, gold nanoparticles can induce their differentiation into osteocytes, a property attributable to such gold nanoparticles [125]. Conversely, carbon nanoparticles can reduce the viability of human MSC cells [4]. Figure 5 shows the most representative cell types and coated used in nanothecnology.

On the other hand, transfection consists of the introduction of foreign genetic material into eukaryotic cells by plasmids, allowing the passage of genetic material (such as supercoiled DNA constructs or siRNA) by electroporation. Additionally, the use of liposomes with mixed lipids allows its fusion with the cell plasma membrane and also enhance its deposition inside the cell [119]. The negative charge on the surface of MSCs and the hydrophobicity allow the drug anchorage [71]. The “anchoring” occurs successfully with silica NPs to the surface of MSCs by specific recognition molecules against cancer cells. In this way, the expression of CD73 and CD90 antibodies on the cell membrane of MSC contributes to the specific binding of silica NPs, ensures a high circulation for long periods, and increases the homing toward the tumor localization [123,124]. 

### 3.1. Interaction between Nanoparticles (NP) and Stem Cells

The preparation of NPs, along with their physicochemical properties, affects the biological function of MSC. Several mechanisms are involved in metallic NP-induced cellular proliferation and differentiation while oxidate reactive species and inflammatory mediators as well as the regulation of several transcription factors are factors that affects the differentiation of MSC in several cell types [72]. 

The silica-coated magnetic NP decreased human hBM-MSC migratory capacity by reducing membrane fluidity and also impairs the expression of focal adhesion molecules [73]. Regarding stem cell differentiation, previous studies have demonstrated positive and negative impacts of AgNPs on stem cell differentiation. As an example, AgNPs of size 10 or 20 and 30 nm are not toxic for MSCs cells but did not affect the differentiation. The shape, size, and surface characteristics of AuNPs impacted their potential to induce the osteogenic differentiation of MSCs. In human adipose-derived stem cells (hADSCs), the induction of osteogenic differentiation is prompted by AuNPs (sized 30 and 50 nm). In addition, gelatin hydrogels loaded with AuNPs increased the proliferation and osteogenic differentiation of hAD-MSCs. In human hBM-MSCs and MC3T3-E1 cells, miR029b-delivered polyethyleneimine (PEI)-capped AuNPs efficiently promoted the osteogenic differentiation without significant toxicity; In addition, L(D)-PAV-AuNPs-exposed MSCs showed upregulation of osteogenic differentiation marker genes; in another study the small size of AuNPs (4 nm) affects the differentiation of hBM-MSCs compared with large size AuNPs (40 nm). The small AuNPs markedly suppressed osteogenic differentiation and induced the adipogenic differentiation of hBM-MSCs. Collectively, the differentiation of stem cells in small size AuNP-treated hBM-MSC is influenced by oxidative stress and inflammation. Interestingly, AuNP-loaded functionalized nanofibrous scaffold promoted the cardiogenic differentiation of the MSCs by upregulating cardiogenic differentiation-related markers [72]. The affinity of MSCs with metallic nanoparticles and polymeric nanoparticles has been studied in cite [124] 

Clathrin or caveolin-dependent endocytosis is considered to be the main mechanism for the uptake of nano-size materials. Clathrin and caveolin-dependent endocytosis, phagocytosis, macropinocytosis, and pinocytosis represent mechanisms for the cellular internalization of NPs. The exocytosis or release of NPs is carried out via vesicle-dependent release, non-vesicle-dependent release, and lysosomal secretion. The treatment with several inhibitors (nocodazole, lovastatin, chlorpromazine, and cytochalasin A) could affect the cellular uptake in vitro and are tools for the study of NP cellular internalization [127,128,129,130,131,132,133].

The interaction of the nanomaterials with the cell membrane or intracellular components, as well as signaling pathways by internalized NPs, were reported in vitro [128,133]. The conjugation of polyethylene glycol to the surface of NPs is one of the key tools for surface modifications of the carriers (see Table 1 and Figure 5). As an example, the AuNP represents a model metallic NP with high efficiency for delivering nucleic acids, recombinant proteins, and drug compounds to the tumor [134]. As an anti-cancer strategy, the delivery of methotrexate, a well-known anti-cancer drug, was conducted by the binding of its carboxyl group with the surface of AuNP [89]. For additional details on the possible application of noble metallic NPs in drug delivery for cancer therapy consult cyte [72,135]. These treatments with AgNPs of 10, 20 and 30 nm size are not toxic for MSCs and as coating material, could increase the adipogenic differentiation of hBM-MSCs and induce the osteogenic differentiation of urine-derived stem cells [128,129,130,131,132,133,134].

### 3.2. Agents for MSC Vehicles

Adaptaded from [124].

Table 2 shows brief information on the size of different NPs [124]).

## 4. Related Problems with the Use of Nanoparticles and MSC for Drug Delivery in Tumor Cells

NPs have been used to treat specific applications, such as site-specific drug delivery systems, and also against tumor growth and more clinical applications [136,137,138,139,140,141,142,143,144,145]. However, the MSC as vehicles for drug delivery with NPs can show inefficient accumulation in tumor sites, possible clearance by reticuloendothelial organs, and toxicity; to solve these problems, strategies aimed at the internalization or conjugation of drug-loaded nanoparticles in MSCs [146]. Different types of stem cells as vehicles like adult stem cells or induced pluripotent SCs (iPS) are also effective. Hypoxic preconditioning of MSCs loaded with PEG-superparamagnetic iron oxide nanoparticles enhanced their migration towards gliomas and their trafficking across the blood–brain barrier [147]. These MSC stromal cells can be useful as anticancer drugs but also depend on the type of cancer, the stage of progression, and its location. On the other hand, interactions between living cells and agents (drugs or nanoparticles) can affect the viability of MSCs and alters pharmacokinetics or pharmacodynamics properties of the drug. For example, plasmids encoding the cytosine deaminase and uracilphosphoribosyl transferase suicide genes has successfully been achieved by polyethyleneimine (PEI) by coating of mesoporous silica nanoparticles on MSCs. These PEI-plasmids-PN induced cell death in breast cancer cells without toxicity in carrier MSCs [148]. Cellular interactions require functional proteins and therefore, if protein–protein interactions on the cell surface of MSCs are destabilized, the homing capacity of MSCs towards injured areas could be reduced [143,149]. Another advantage that allows the use of MSCs as carriers of drugs in cancer is the low risk of malignant transformation [128]. In addition, MSCs are resistant to the cytotoxic effects of paclitaxel [133], and paclitaxel exposure does not affect the expression of the transmembrane pump P-glycoprotein 1 (gp-P) of MSCs, a mechanism that provoke resistance to paclitaxel in tumoral cells [134] while the drug doxorubicin (DOX) was successfully introduced into MSCs cells [129]. Covalent conjugation or physical association of NPs to the surface of MSCs can significantly increase drug load delivery by endocytosis. On the other hand, doxorubicin at clinically used doses promotes premature senescence of MSCs in vitro [132]. The encapsulation of chemotherapeutic drugs in NPs increases the drug loading capacity, and also guarantee the delivery of a dose therapeutic anticancer drug at the tumor site [131]. For example, ultrasound-sensitive mesoporous silica-loaded MSCs selectively release charge when stimulus is applied, both in vivo and in vitro [148]. When loaded with doxorubicin, the NPs kill breast cancer cells in vitro by ultrasound. The sustained release of PLGA nanoparticle-encapsulated paclitaxel does not affect the functional capacities of MSCs or their tumor tropism and also decreases glioma tumor tissue [130]. The main advantage of the use of MSCs is its uniformity and good infiltration within the tumor [9,150]. Paclitaxel-loaded poly-(co-glycolic acid, PLGA) nanoparticles exhibited dose-dependent cytotoxicity in lung cancer cells both in vivo and in vitro models [81]; the paclitaxel-NPs-MSCs platform accumulated in lung tumors and enhanced antitumor effects [76]. The combination of NPs in MSCs decreases tumor growth survival with less systemic toxicity that nanoparticle-encapsulated drug treatments in a variety of tumors [77]. Additionally, to the cancer drugs employed, the NPs could also serve as carriers for bioactive molecules (e.g., DNA, mRNA), in MSCs. NPs-based anti-angiogenic systems have demonstrated antiangiogenic effects of various types of NPs in rodent models (gold, silica, silicates, diamond, silver, and copper) [80,82]. For example, glycolic acid polyconjugates enhance NP drug delivery in human MSCs [114]. MSCs loaded with paclitaxel-PLGA nanoparticles are selectively accumulated in the lungs of tumor-bearing mice. For example, paclitaxel is known to blocks tumor cell proliferation by inhibiting angiogenesis [83] but provoques haematological toxicity (e.g., leukopenia and neutropenia) as well as liver damage. This system maintains the plasma paclitaxel concentration for a longer period of time. Finally, nanoengineered of MSCs significantly reduced tumor cell proliferation, decreased angiogenesis, and leukopenia was less severe due to the very low dose of paclitaxel [151]. The conjugation of different types of NPs to MSC ensures the success of the feasibility of tracking migration toward the tumor size by non-invasive image techniques in preclinical and clinical settings, such as magnetic resonance imaging (MRI), computed tomography (CT), ultrasound, optical imaging, positron emission tomography (PET) and single photon emission computed tomography (SPECT). However, many aspects remain to be resolved regarding its biosafety and efficacy, such as the long-term stability of NPs combined with MSCs and the drug release capacity in cancer patients to ensure its clinical efficacy. Adipose-derived MSCs loaded with manganese oxide coated on mesoporous silica have been shown to be more effectively detected by MRI imaging for long periods after its transplantation [108] because the number of MSC cells can be visualized in the tumor site (see Figure 6). 

The Table 3 summarizes (in general) the advantages and problems of Nanoparticles (NP).

### Is It Possible to Use Exosomes (exos) as Carrier of Drugs?

Exosomes could be natural carriers of therapeutic agents for cancer therapy because they are small extracellular vesicles (diameter: 30–150 nm) released from cells after the fusion of an intermediate endocytic compartment, the multivesicular body (MVB) with the plasma membrane; they could transfer of macromolecules such as lipids, proteins and nucleic acids (mRNA, tRNA, non-coding RNA, microRNAs, and mitochondrial DNA). Compared to synthetic nanomaterials, a first advantage of using exos is the innate biocompatibility and low immunogenic/cytotoxic effect that it provokes in cells. Furthermore, exosomes cross the blood–brain barrier, avoid rapid clearance by the kidneys and bypass the mononuclear phagocytic system. The fate of exos can be labeled with detectable markers [152,153,154,155,156]. To optimize the efficacy of MSC exoderivates for drug delivery, the optimal dose, timing, and route of administration must be monitored to achieve maximum efficacy [109]. The advantages and disadvantages of its use depend on the sensitivity, specificity, penetration, radiation, and spatial resolution. Although many types of cells produce exos, MSCs are more prolific and, therefore, suitable for drug delivery [155]. MSC exoderivates have intrinsic capabilities similar to those of MSCs and can penetrate the tumor site [111]. It is known that exos from MSC carry specific mi-RNAs that could inhibit tumor growth [37]. The exosomal lipids induce human pancreatic tumor MiaPaCa-2 cell resistance via CXCR4/SDF1 alpha signaling pathways [157].

Previous studies have shown that intracellular delivery of siRNA to knockdown CXCR4 expression could be a therapeutic strategy in cancer cells. For example, nanoparticles (NPs) using polylactic-co-glycolic acid (PLGA) loaded with the secretome from MSCs to form MSC-Sec NPs are protective. In this study, MSC-Sec/CXCR4 NP was accumulated in bone and also inhibited the osteoclast differentiation while inducing osteogenic proliferation, suggesting that MSC-Sec/CXCR4 NPs reduce ovariectomized-induced bone mass rats [158,159].

The antitumor activity of the nanodrug consisting of doxorubicin and exosome (Exosomes-Doxorubicin: Exo-Dox) derived from MSC has been evaluated in vitro and in vivo. For this aim, the exosomes were isolated with a Kit (Exo-Dox) and examined by NP tracking analysis (NTA) as well as transmission electron microscopy. In this study, MSC-derived Exo-Dox as nanocarriers release the chemotherapeutic drug against osteosarcoma proliferation, SDF1-CXCR4 signaling the cur factor of proliferation [39].

In another study, the fused CXCR4^+^ exosomes with liposomes carrying antagomir-188 produced hybrid NPs. This hybrid NPs specifically gathered in the bone marrow and released antagomir-188, which reversed age-related trabecular bone loss and also decreased cortical bone porosity in mice [115].

Finally, clinical trials of MSC exoderivates are currently in progress by gene delivery and also evaluate its immunomodulatory properties [40].

## 5. Nanoparticles as si-RNA CXCR4 Carriers for Preventing Tumor Growth

The main strategy of this review is to discuss the role of CXCR4 blockade against cancer by attacking MSC toward the tumor cells; however, some of the cancers that initially respond to chemotherapy become resistant to treatment, and stem cells could be vehicles for drug-loaded NPs seems to be a very promising strategy to target tumor tissues [87]; the need of new therapies against cancer could reduce the toxicity of NPs carriers of drugs in and NP-based biomimetic systems could provide an important solution for drug delivery, although some physical and chemical modifications could decrease non-specific interactions for many biomedical purposes.

The use of NP that overexpress CXCR4 facilitate the homing in injured tissues or tumor sites [160]. Genetic engineering, thanks to viral vectors, allows us to overexpress certain proteins capable of enhancing homing and the drug-vehicle capacity of MSCs towards the tumor cell; however, the occurrence of mutations and the possible activation of oncogenes limits its clinical use [43]. For example, overexpression of the CXCR4 on the surface of MSC cells clearly increases their ability to migrate toward tumor sites by SDF1 alpha release [161]. Another study proposed genetically engineered mesenchymal stem cell membrane-derived NP with the active targeting capability. BM-MSCs were engineered by overexpression of CXCR4 to actively migrate toward the tumor sites by identification of fluorescent NPs. The modified NPs loaded with the therapeutic drug increased cellular uptake by a better penetrance in the target cell [162].

On the other hand, the transfection of the CXCR4 receptor by adenovirus or lentivirus enhances their efficacy as drug-carrying vehicles and/or promotes repair in several models of injury; in fact, the adenovirus transfection of CXCR4 protects against myocardial infarction [4]. In order to avoid the potential risk of viral transduction, the use of lipids, polymers, and inorganic NPS is promising [163,164,165]. However, the lack of membrane integrity, as well as mitochondrial alterations, limit its use and also affect its capacity as carrier of human MSCs [166,167].

The SDF1/CXCR4 axis, associated with cancer stem-like cells, is a target of cancer cells and can be overexpressed under pathological conditions; SDF1 is highly enriched in the bone marrow for CXCR4-positive hematopoietic stem cell (HSC) and also promotes tumor bone metastasis. In fact, CXCR4 is a promising therapeutic target against cancer since the intracellular siRNA delivery of its knockdown can prevent tumor growth. The radiation for glioblastoma (GBM) treatment can provoke systemic toxicity and provokes blood–brain barrier (BBA) disruption, thus, allowing glioblastoma (GBM) to cross BBA. This aggressive brain cancer tumor promotes proliferation by CXCR4/SDF 1 alpha chemokines. Although the CXCR4 antagonist (AMD3100) could be an attractive anti-GBM therapeutic target, its poor pharmacokinetic properties, and unfavorable bioavailability, limits its clinical use. Thus, new synthetic protein NPs (SPNPs) coated with the transcytotic peptide iRGD (AMD3100-SPNPs) could abolish CXCR4/SDF-1 alpha overexpression in GBM. These AMD3100-SPNPs treatments blocked the CXCR4/SDF1 alpha pathway in human GBM models in vivo and rodent models of cancer. Thus, the CXCR4 blockade by NPs abolished GBM proliferation, decreased infiltration of CXCR4^+^ monocytic myeloid-derived suppressor cells (M-MDSCs) into the tumor, and restored BBA integrity. Collectively, the combination of AMD3100-SPNPs plus radiation led to long-term survival by preventing tumor recurrence without additional treatment [167]. Thus, CXCR4 is a promising therapeutic target because the intracellular delivery of siRNA (knockdown) or CXCR4 blockade by antagonist can prevent tumor progression in mice. Thus, the intracerebral infusion of nanocarriers plus CXCR4-radiopharmaceuticals could have clinical applications [168].

On the other hand, CXCR4 turnover is mainly regulated by autophagy; a recent study found that SiO_2_-NP internalization augmented CXCR4 surface levels while bafilomycin-A1 (an autophagy inhibitor) reproduced CXCR4 overexpression in control hMSCs. These chemotaxis assays demonstrated that SiO_2_-NPs enhanced hMSC migration toward SDF 1 alpha levels. Collectively, SiO_2_-NP internalization induces MCS chemotaxis via CXCR4/SDF1 alpha signaling in vitro [169,170].

Another study with CXCR4-targeted lipid-calcium-phosphate NPs with nitric oxide (NO) donors (LCP-NO NPs) demonstrated that delivery of NO enhanced BBA permeability and also improved gene delivery across the BBB. In this study, the codelivery of NO and PD-L1 siRNA by CXCR4-targeted NPs could be immunotherapy against GBM tumorigenesis [161].

The effective transport of small interfering RNAs (siRNAs) can be induced with hyaluronic acid receptors and also carriers with low-molecular-weight polyethyleneimine (PEI)-based transport systems. Gold nanoparticles (AuNPs) and their conjugates with PEI and HA ranged from 25 to 690 nm, and CXCR4 gene silencing on the MDA-MB-231 cell line drastically decreased breast cancer progression [171].

On the other hand, Zinc oxide nanoparticles reduced tumor growth but induced overexpression of cytochrome P450; the effectiveness of zinc oxide nanoparticles via doping with lanthanides, such as samarium, showed more antitumor activity than other lanthanides by downregulating CXCR4 and regulating PI3K/Akt/signaling pathways in Ehrlich solid tumor with minimal toxicity [172].

At the clinical level, the CXCR4 overexpression has been reported in women with advanced endometrial cancer (EC); the antitumor activity of two CXCR4-targeted NPs in vitro, including either the *C. diphtheriae* (T22-DITOX-H6) or *P. aeruginosa* (T22-PE24-H6) toxin were tested by repeated subcutaneous administrated of both nanotoxins in an EC mouse model these nanotoxin-induced antitumoral effects were observed in vitro as well as in vivo by inducing apoptosis without any off-target toxicity. In addition, repeated T22-DITOX-H6 administration in the metastatic model drastically decreased tumor burden while significantly blocking peritoneal, lung and liver metastasis; thus, both nanotoxins, and especially T22-DITOX-H6, are promising therapeutic drugs against this type of cancer that overexpress CXCR chemokine receptor [173,174,175].

The therapy using CXCR4 ligands, such as DV1 peptide, on drug-loaded NPs is a potential strategy for cancer prevention. Another study evaluated the role of SDF-1 alpha on magnetic nanoparticle delivery system for stem cell recruitment by creating a purified biotin-labeled SDF-1 alpha protein, which was immobilized on streptavidin-modified magnetic nanoparticles (MNP) and also enhanced by streptavidin-biotin linkage; in this study, SDF-MNP showed cytotoxicity even at a concentration of 125 µg/mL MNP although promoted chemotaxis of MSC in vitro and in vivo.

The antitumor effect of the CXCR4-targeting drug delivery system has been evaluated in U87MG cells (express CXCR4) by using avidin-poly (lactic-co-glycolic acid, PLGA) NP surface tagged with biotinylated DV1 peptide ligand. For this purpose, a double-emulsion solvent evaporation technique was used in order to prepare avidin-PLGA nanoparticles, which were characterized by transmission electron microscopy. The uptake was evaluated by confocal microscopy after incorporating fluorescein isothiocyanate (FITC)-labeled albumin inside these NPs; the avidin-PLGA NPs were successfully synthesized and verified by FITC-labeled biotin staining in vitro. As a whole, avidin-PLGA nanoparticle surface tagged with biotinylated DV1 peptide ligand is a strategy for preventing tumor progression in CXCR4-expressing cancer cells [132]. As a whole, these findings suggest that SDF-MNP can promote bone repair by stem cell-based therapies [176].

It has been demonstrated that CXCR4 targeted polymeric nanoparticles (PGLA) (LFC131-NPs) and PLGA NPs encapsulating DOX (LFC131-DOX-NPs). In this study, LFC131-NPs blocked SDF-1 alpha-induced migration of BT-549-Luc cells [177], while LFC131-NPs and LFC131-DOX-NPs decreased cell viability in a dose-dependent manner [178].

In another study, SPIONs with bioorthogonal azide and alkyne surfaces masked by polyethylene glycol (PEG) layers tethered to CXCR4-targeted peptide ligands were synthesized and characterized. The simultaneous systemic administration of the bioorthogonal SPIONs in tumor-bearing mice demonstrated the signal-enhancing ability of these ‘smart’ self-assembling nanomaterials for preventing tumor growth (positive for CXCR4 and MMP2/9 metalloproteases [178].

Another study has explored the translocation domain of diphtheria toxin (TD: blocks CXCR4 downstream signaling pathways) [179], as a functional domain in CXCR4-targeted oligomeric nanoparticles designed for cancer treatment. This TD nanoparticles partially abolished the receptor specificity, and they induced nonspecific internalization in mammalian cells [180].

Since therapies for metastatic colorectal cancer have not yet been solved, the use of NPs could be a better therapeutic strategy than chemotherapy and immunotherapy. Preclinical findings support the safety of 177Lu-PLGA(RGF)-CXCR4L as a potential combined treatment against colon cancer. In this study, empty PLGA and PLGA(RGF) NPs were generated, and these 177Lu-PLGA(RGF)-CXCR4L NPs decreased cell viability and reduced the tumor size by inducing apoptosis of HCT116 colorectal cancer cells in a xenograft model via inhibiting Erk and Akt phosphorylation [180].

## 6. Conclusions and Future Perspectives

The MSC-based systemic cell therapy seems to be a good strategy for the treatment of aggressive tumors. NPs act as delivery carriers of certain anticancer agents [181] and other applications. Thus, exploring the tropism of MSCs towards tumor sites to deliver gene therapy and NP technology are promising strategies to eradicate tumors. On the other hand, the transfection of CXCR4 in cell cultures of MSCs enhances the homing and promotes repair effects by transplanted stem cells [12]. MSCs can be genetically modified, and these MSCs express antiproliferative, proapoptotic, or antiangiogenic markers to treat different types of tumors; these NPs are carriers of drugs, although some physical and chemical modifications could reduce non-specific interactions. For example, adipose-derived MSCs (AD-MSC) loaded with manganese oxide coated on mesoporous silica have been shown to be more effectively detected by MRI imaging in the tumor sites. For stem cell-based therapies, the fate and distribution of stem cells should be traced by using non-invasive methods and nanomaterial-based labeling agents [73]. However, evaluation of the biophysical effects and related biological functions of nanomaterials in stem cells must be improved before the clinical detection of tumors CXCR4-dependent levels. Thus, the optimal NP concentrations to preserve MSC migratory are necessary for a successful translation to humans, although more studies are necessary that confirm the safety and the absence of adverse effects.

Certain carriers like glycolic acid polyconjugates enhance nanoparticle drug delivery in human MSCs [114]. In addition, a genetically modified NP functionalization strategy could prevent tumoral progression [123]. The si-CXCR4 delivery in stem cells is a promising option for preventing tumoral growth. However, several problems in terms of stem cell delivery/recruitment efficiency toward sites of tumors should be improved by the combination of NPs and MSCs in order to decrease possible toxic effects and also improve the clinical efficacy of these si-RNA for CXCR4 by increasing the apoptosis of tumor cells with less systemic toxicity that NP-encapsulated drug treatments [27,182]. Finally, MSC exoderivates are an optimal treatment for cancer, given their strong penetrance in the tumor [183].

Recently, extensive research efforts have been directed towards the application of nanotechnology in stem cell research, and the search for new nanomaterials with better physical and chemical properties could facilitate stem cell differentiation and proliferation. Of note, we must understand how metallic NPs affect stem cell differentiation and the effects of dose exposure and time of exposure; however, certain NPs induce differentiation, but others suppress differentiation. Thus, further research will find the optimal NP conditions required for an appropriate differentiation of stem cells. In addition, we should better understand how metallic NP-based regulates stem cell function in more detail; more importantly, we must understand contradictory findings on the effects of certain metallic NPs on stem cell differentiation, and more studies with NP-MSC should be performed in rodent models of disease before its clinical application. As an example, the understanding of molecular mechanisms by which iPSC reprogramming and differentiation take place on metallic NPs will add particular information on further clinical applications against tumors. We must also reduce the toxicity of metallic NPs on stem cells. Unfortunately, several metallic NPs affect negatively the impact on stem cell proliferation and differentiation, which is a concern that should not be avoided in patients. Future studies on metallic NP-related stem cell toxicity are required. In addition, reports on the impact of SiNPs in stem cell differentiation are limited; the studies on the toxicity of SPIONs in stem cells are obscure and need further elucidate. In addition, new methods such as flow cytometry could provide information about thousands of cells (events) [183,184,185,186] with the very sensitive method in one sample after determining the average fluorescence intensity per particle; however, the inconvenience of this method is that does not distinguish between intracellular and membrane-associated NPs [72]. Thus, novel hybrids of the metallic NPs with other nanomaterials need to be developed for promoting the high-quality stem cell differentiation. However, further efforts are needed to find the best design and optimal exposure conditions of metallic NPs for a good level of stem cell differentiation without toxicity. Finally, drug discovery could identify new chemokine blockers for clinical applications with strong antitumoral activity and better pharmacodynamic properties without adverse effects on the patient. As combined strategy, chemoattractants as CXCR4 and epidermal growth factor receptor (EGFR) can recruit DEX-SPION labeled hMSC to the site of damage [187]. Thus, these combined strategies could improve the efficacy. The introduction of CRISPR–Cas technology to specifically target key components of endocytic machinery could increase the NPs uptake mechanisms. The creation of a better library of potential molecules (natural and synthetic) that are either stimulative (vitamins, polysaccharides, inflammatory mediators, toxins) or reducing NP uptake should be considered since nano–bio interactions as well as internalization increase the efficacy of nanomedicines. In terms of stem cell therapy, several factors should be used to increase the survival rate or avoid the differentiation into undesired lineages. Moreover, better labeling agent to track MSCs or gene/drug delivery carriers could enhance the clinical efficacy of MSCs. However, since NPs can induce cell death, inhibit cell proliferation or influence the differentiation of MSCs, the preparation of multifunctional NPs in conjunction with a better cellular internalization of the NPs could increase the clinical efficacy of nanotechnology in stem cell-based therapy. In this way, cell membrane-coated nanoparticles could increase the delivery and also evade the immune system response. As an example, a study with two mesenchymal stem cell (MSC) membrane-coated silica nanoparticles (MCSNs) with similar sizes but different stiffness values (MPa and GPa) showed unexpected results. In fact, a much lower macrophage uptake, but much higher cancer cell uptake, was found with the soft MCSNs compared with the stiff MCSNs. Thus, the study of soft MCSNs could better understand these effects on MSC since they reported the formation of a more protein-rich membrane coating rich in CXCR4 and MSC surface marker CD90. This led to the soft MCSNs enhancing cancer cell uptake mediated by CXCR4/SDF-1 chemokines. Thus, are necessary more studies on bio-nano interactions by evaluating factors such as elasticity; finally, the study of optimal NP concentrations to preserve stem cell migration toward tumor sites without adverse effects are crucial aspect before its clinical application. These strategies should be studied in more detail by evaluating the toxicity and homing of MSC. The design of a new si-RNA for CXCR4 with better BBA penetrance could increase the real capacity for preventing tumor proliferation in the brain. Additionally, for stem cell-based therapies, the fate and distribution of stem cells should be label in a nanomaterial agent and detected by non-invasive methods. In fact, more studies with NP-MSC in rodent models of cancer confirm the low toxicity before its clinical application. However, evaluation of the biophysical effects and related biological functions of nanomaterials needs to find NPs that do not affect cell viability or membrane fluidity of MSCs. The research should include better NP with higher internalization capacity without toxic effects and also the ability to reduce intracellular ROS and peroxidation production; finally, the study of new biophysical properties and better migratory capacity (homing) of NP-MSC could generate NPs with better cell trafficking.

## Figures and Tables

**Figure 1 micromachines-14-02068-f001:**
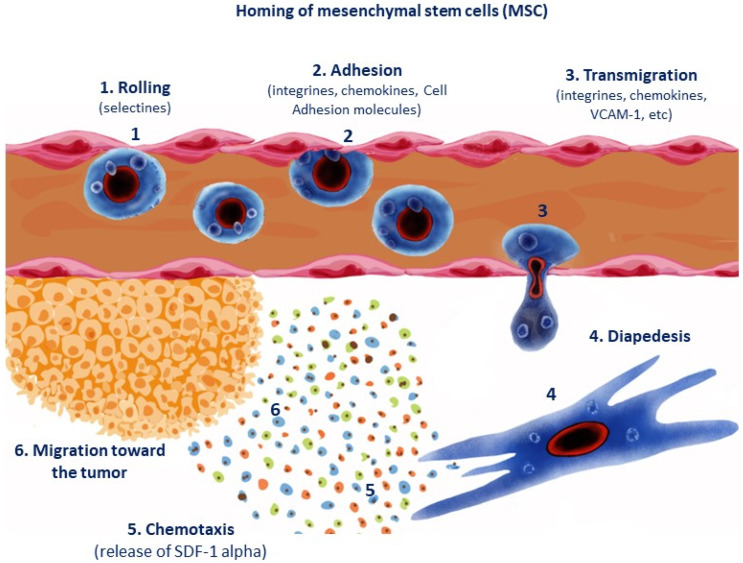
Homing of mesenchymal stem cells (MSCs) toward the tumor site is regulated by CXCR4/SDF1 alpha levels. Inflammation is a calling signal for CXCR4 receptor-bearing MSCs and enhances the recruitment of MSCs [1].

**Figure 2 micromachines-14-02068-f002:**
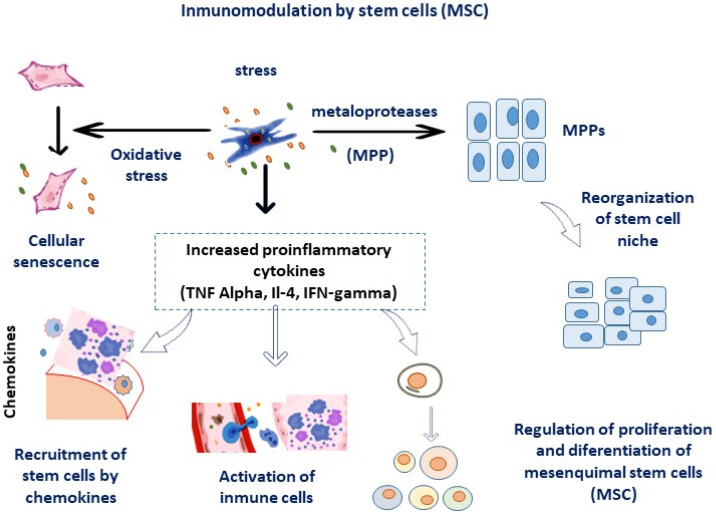
Immunodulatory effects of mesenchymal SCs (MSCs).

**Figure 3 micromachines-14-02068-f003:**
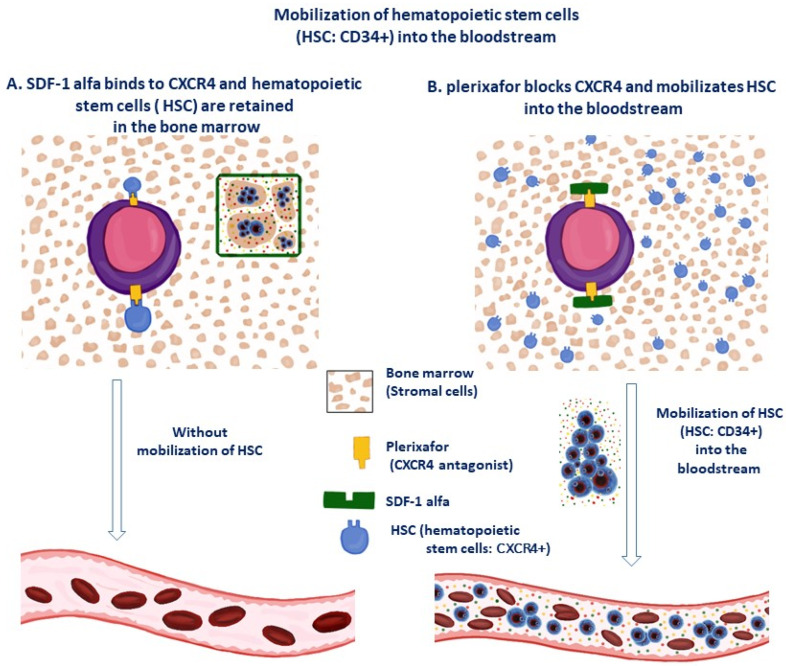
Mobilization of hematopoietic SCs (HSCs) by drugs (plerixaform, AMD-3100, CSF, or even viagre).

**Figure 4 micromachines-14-02068-f004:**
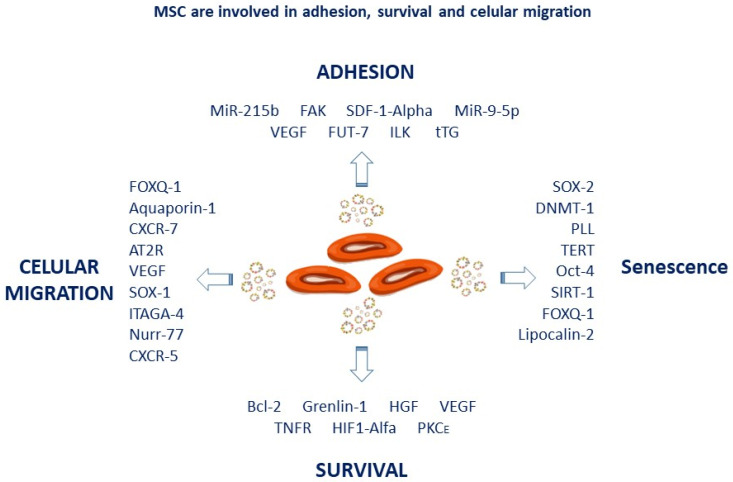
Trophic factors released by MSCs contribute to cell adhesion, migration, survival, and senescence. For example, CXCR4 or CXCR5 chemokines increase MSC mobilization, while trophic factors such as HGF VGEF play a role in survival.

**Figure 5 micromachines-14-02068-f005:**
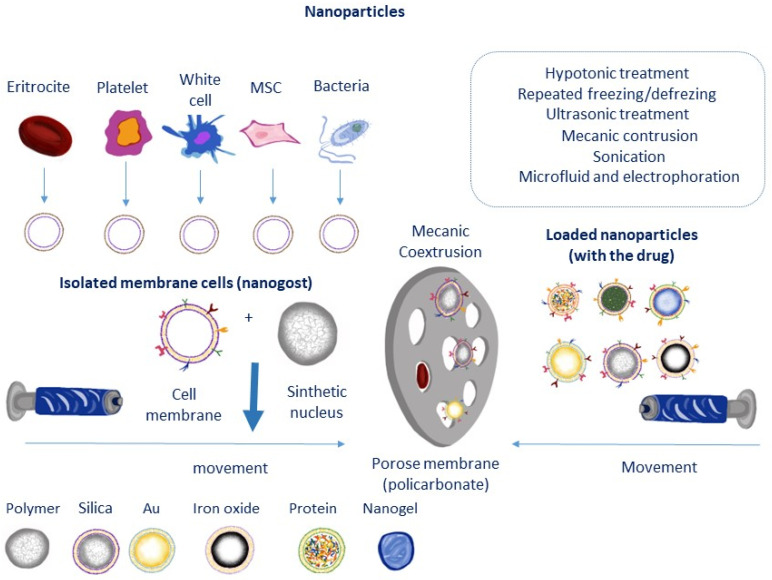
Nanoparticle technology and physical methods for the introduction of several carriers in different types of cells (MSC, eritrocytes, etc…). Adapted from [87].

**Figure 6 micromachines-14-02068-f006:**
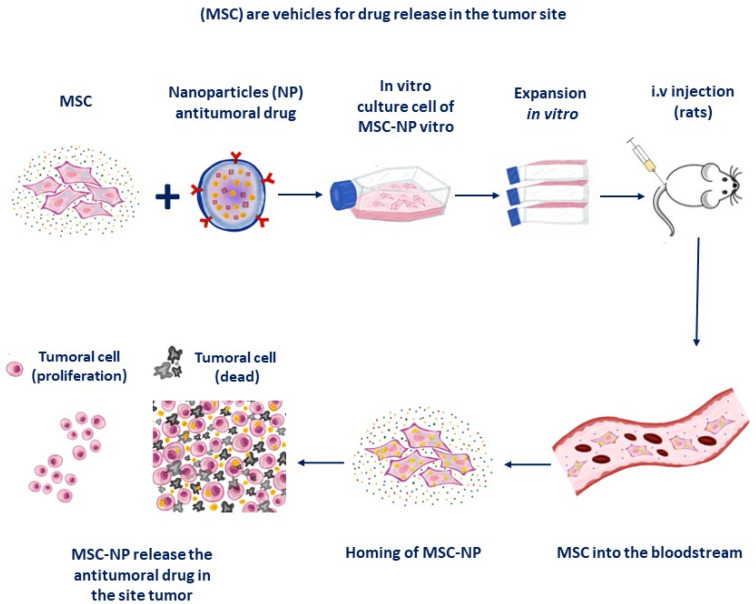
MSCs with antitumor drugs can eradicate a tumor using nanoparticle technology. Circulating MSCs can be targeted to the tumor by gene therapy when they overexpress the CXCR4 receptor by i.v injection in a rodent model of cancer, whereby tumor-reaching cells attempt to eradicate the tumor once released (Adapted from Pereboeva et al. [124]).

**Table 1 micromachines-14-02068-t001:** Mechanisms and agents for the detection of tumors in tissues.

**Receptor-ligand binding**	PTX Gencitabine Gadolinium Nanoparticles (load with cumarines)	Leukemia Pancreas cancer Diagnosis of cancer Glioma
**Transduction**	mR-133	Miocardic infarct (acute)
**Transfection**	TRAILs	Lung metastases
	HIF-1 Alpha	Miocardic infarct (acute)

**Table 2 micromachines-14-02068-t002:** Nanoscale unit, major compound, and shape/size.

Nanoscale Unit	Major Compound	Shape and Size
Iron oxide	Fe_3_O_4_	Spheroidal, rods, cubes, hexagonal 5–10 nm size-dependent emission
	Iron (III)	(fluorescence probe)
Iron (III) oxide-hydroxide	Iron (III) oxide-hydroxide	2–3 nm size Spherical, hexagonal, rods, cubes
ZnO NS	Zn	The ZnO-NPs are pseudo-spherical forms with an average particle size between 11 and 20 nm The average size 28 nm Very short hexagonal rods, hexagonal wurtzite structure, globular shaped particle-like structures, a mix of wide slates and thin hexagonal rods, thin slates-like structures, thorn-like morphology with wurtzite crystal structure, highly crystalline, having wurtzite crystal structure, spherical in shape with a smooth surface
Zn-Capped	Cd, Se	1–6 nm size-dependent visible absorption/scattering (contrast)
zinc oxide nanoparticles with PEG	Zinc oxide PEG	The prepared ZnO-NPs had a hexagonal shape and average particle size of 20–40 nm,
ZnO NPs obtained by using cellulose derivatives as macromolecule	ZnO (cellulose)	Zn NPs clusters (20 nm to 240 nm of size) Average size: 24 to 40 nm Wurtzite orhexagonal structure, nanowires, Rod-like and plate-like crystals, nanorods Spherical (Carboxymethyl cellulose) capped Ag-ZnO nanoparticles) Spherical (Hydroxyethyl Cellulose). Rod-shaped (ZnO-overlaid cellulose nanocrystals) Hexagonal wurtzite structure (Cellulose–ZnO-hybrid nanocomposite)
Au (AuNPs)	gold	Shape: spherical Single spherical gold nanoparticles with an average size of 5.7 nm and coral-shaped; the shape of AuNPs in these solutions was mainly spherical.
Au nanorod	Gold	The shape of non-spherical gold nanoparticles are rods, wires, cubes, nanocages, (multi-) concentric shells, triangular prisms, and other more exotic structures (such as hollow tubes, capsules, and even branched nanocrystals). Au NP have several sizes 5–10 nm 10–100 nm length, dependent visible absorption/scattering magnetic resonance (RMN)
Aluminum (Al) oxide NP	Al	Shapes: Au nanorods with spherical heads, Dogbone-like Au nanorods, dumbbell-shaped Au nanorods, cuboidal shape. Size: Al nanopowder particles typically ranging in size from 10 nm to 5 μm The different shapes of Al_2_O_3_ nanoparticles that are under contemplation are column, sphere, hexahedron, tetrahedron, and lamina.
Ag NP	Ag	2–100 nm size-dependent magnetic properties (MR contrast agent, detection probe). Shapes: spheres (diameter 40–80 and 120–180 nm; cubes (140–180 nm), and rods (diameter 80–120 nm, length > 1000 nm).
Dye-droped silica	silica	10–100 nm size-dependent magnetic properties (fluorescence probe) The silica nanoparticles are spherical, uniformly dense, and sizes are ∼ (50–60) nm inside the performing matrix. The organic laser dye molecules are attached using electrostatic interaction on the surface of silica NP (fluorescence).
Polimeric micelle Au	Polyeric micelles prepared from amphiphilic block (di- or tri-) or from graft copolymers.	Typical polymeric micelles have a spherical shape (cubical and hexagonal), and the size is between 10–100 nm. For example, can be used as detection probe (10–20 nm size-dependent dispersion stability)
SPIO magnetic iron oxide Nanoparticles	magnetic iron oxide NP	These NP are classified according to their size as ultrasmall superparamagnetic particles of iron oxide (20–50 nm diameter), superparamagnetic particles of iron oxide (SPIO) (60 to ∼250 nm), and microparticles of iron oxide (MPIO) (from 0.9 µm upward).
(Fe_3_O_4_-NP)	Tetraoxide of iron	Shepres (216.6 nm), cubes (158.5 nm), octaaedral (4.9 nm).
Carbon nanoparticles	carbone	1–10 nm of length-dependent solvent extraction (cell membrane translocator) Nano to micrometer length Single-walled carbon nanotubes have diameters around 0.5–2.0 nm. Shape: types of carbonaceous NPs with lamellar/flaky shape, spherical shape, and tubular shape.

**Table 3 micromachines-14-02068-t003:** Advantages and problems of nanoparticles.

Advantages	Problems
-Nanotechnology allows procedures, biomedical applications, including cancer treatment	-Toxicity -Low efficacy -Low homing of stem cells and the lack of interaction with NP
-Development of more effective drug delivery for si-CXCR4 release in the tumor site -Nanotechnology can help with creating what is called smart drugs. -NP allows the use of nanotubes, aerogels, nano particles with stronger, more durable, and better psychicochemical properties -Nanotechnology as vehicle of antitumoral drugs into the site tumor -NP allows a better immune regulation.	-NP nanotechnology is very expensive and difficult to manufacture -It is necessary to use better tracers for medical image technology (fluorescence probes) -Development of trackers with better staining in the tissue are required. -Better molecular techniques for si-RNA CXCR4 delivery with less toxic effects and better penetrance in the tumor

## Data Availability

No applicable.

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
