# Peer review of "Nanoparticles and Mesenchymal Stem Cell (MSC) Therapy for Cancer Treatment: Focus on Nanocarriers and a si-RNA CXCR4 Chemokine Blocker as Strategies for Tumor Eradication In Vitro and In Vivo"

_micromachines, 2023, doi:10.3390/mi14112068_

Round 1

Reviewer 1 Report

Dear Editor of Micromachines

This MS is described the using nanoparticles and mesemchymal stem cell (MSC) therapy for cancer treatment and focused on nanocarriers and si-RNA chemokine 3 CXCR4 blockade as strategies for tumor eradication in vitro an in vivo. This is an interesting study; however, some points need to be addressed.

·       It is better to mention the advantages and disadvantages of this approach in a Table.

·       The performance of this strategy is better to be shown in a Fig according to the contents mentioned in the article.

·       According to these mentioned contents, which nanoparticles are suitable for this strategy?

·       There are also some grammatical, typographical and formatting defects in the manuscript. Please correct.

Dear Editor of Micromachines

This MS is described the using nanoparticles and mesemchymal stem cell (MSC) therapy for cancer treatment and focused on nanocarriers and si-RNA chemokine 3 CXCR4 blockade as strategies for tumor eradication in vitro an in vivo. This is an interesting study; however, some points need to be addressed.

·       It is better to mention the advantages and disadvantages of this approach in a Table.

·       The performance of this strategy is better to be shown in a Fig according to the contents mentioned in the article.

·       According to these mentioned contents, which nanoparticles are suitable for this strategy?

·       There are also some grammatical, typographical and formatting defects in the manuscript. Please correct.

Author Response

This MS is described the using nanoparticles and mesemchymal stem cell (MSC) therapy for cancer treatment and focused on nanocarriers and si-RNA chemokine 3 CXCR4 blockade as strategies for tumor eradication in vitro an in vivo. This is an interesting study; however, some points need to be addressed.

  • It is better to mention the advantages and disadvantages of this approach in a Table.
  • The performance of this strategy is better to be shown in a Fig according to the contents mentioned in the article.

Thanks for your comments, which help us to improve the manuscript. We have included a general table on advantages and disadvantahges of Nanoparticles in this R1 version. Thanks again

Advantages of NP (in general).

-Nanotechnology allows procedures, biomedical applications, including cancer treatment

Development of more effective drug delivery for si-CXCR4 release in the tumor site

-Nanotechnology can help with creating what is called smart drugs.

-Nanotechnology allow the use of nanotubes, aerogels, nano particles with stronger, more durable, and better psychicochemical properties

-Nanotecnology as vehicle of drugs allow reléase antitumoral drugs in the site tumor

-NP allow a better inmune regulation.

Disavantages of NP

Toxicity

Low efficacy

Low homing of stem cells and lack of interaction with NP

-NP nanotechnology are very expensive and also is difficult to manufacture

-It is neccesary to improve better tracers for medical image thecnology

-Development of trackers or better markers in the tissue by NP are neccesary

-Better molecular thecnics for si-RNA CXCR4 delivery are required for optimal clinical efficacy.

  • According to these mentioned contents, which nanoparticles are suitable for this strategy?

The type of NP selected depend of the type of tumor. For example, papers on NP (Au, Silica) and nanogels are availables in vitro studies.

  • There are also some grammatical, typographical and formatting defects in the manuscript. Please correct.

We have corrected it. The R1 versioin has been improved by a native english speaker.

Comments on the Quality of English Language

Dear Editor of Micromachines

This MS is described the using nanoparticles and mesemchymal stem cell (MSC) therapy for cancer treatment and focused on nanocarriers and si-RNA chemokine 3 CXCR4 blockade as strategies for tumor eradication in vitro an in vivo. This is an interesting study; however, some points need to be addressed.

  • It is better to mention the advantages and disadvantages of this approach in a Table.
  • The performance of this strategy is better to be shown in a Fig according to the contents mentioned in the article.
  • According to these mentioned contents, which nanoparticles are suitable for this strategy?
  • There are also some grammatical, typographical and formatting defects in the manuscript. Please correct.

We have reply you before. Thanks a lot for your comments¡

Reviewer 2 Report

This an excellent review concerning a topic of great potential importance and will be a great help both to those already in the field and those who wish to enter. It is well organized and presented; There are only some small typo problems here and there that should be resolved:

for example

the title of Figure is incorrect: Movilazation should be Mobliization

line 148: 'figure' is written twice

line 149: please remove the accent on o in the word 'adhesion'

line 210: in the phrase "carry" out that you have used please use just carry

title of Figure 6: write 'tumor cells' and not 'tumoral cells'

and there are some others here and there

In general the English is excellent with just a few errors or typos as outlined above

Author Response

This an excellent review concerning a topic of great potential importance and will be a great help both to those already in the field and those who wish to enter. It is well organized and presented; There are only some small typo problems here and there that should be resolved:

Dear reviewer

Thanks a lot¡ for your comments, which help us to improve the R1 manuscript content. Please, take into account that you have included parts in red color after Reading comments of the another reviewer.

for example

the title of Figure is incorrect: Movilazation should be Mobliization

We have changed it. Mobilization and plerixafor are correct now.

line 148: 'figure' is written twice

Thanks¡. We have corrected now.

line 149: please remove the accent on o in the word 'adhesion'

Done it¡. Thanks¡

line 210: in the phrase "carry" out that you have used please use just carry

Done it¡. Thanks¡

title of Figure 6: write 'tumor cells' and not 'tumoral cells'

Done it¡. Thanks¡

and there are some others here and there

We have corrected all tipo errors.

Reviewer 3 Report

Dear Authors,

The manuscript *Nanoparticles and mesemchymal stem cell (MSC) teraphy for cancer treatment: focus on nanocarriers and si-RNA chemokine CXCR4 blockade as strategies for tumor erradication in vitro an in vivo* is a good theme to be considered for a review. 

Some significant advances from Mesenchymal stem cells (MSCs) in combination with nanoparticles were shown to decrease the tumor growth in vitro and in vivo have been in this work. This manuscript should be improved to be considered in Micromachines, MDPI Journal.

1.        The manuscript has several typing errors along all the text. Please try to make all corrections from the title to the end of this work.

2.        Almost all figures have the same features with errors in typewriting.

3.        Covalent conjugation or physical association of nanoparticles to the surface of MSCs can significantly increase the drug load delivery by endocytosis. This is the main key of the discussion in this work. Some bonding mechanisms between MSCs and Nanoparticles should be considered in the text like -OH,-COOH, etc. What is the binding mechanism affinity of MSCs with metallic nanoparticles and polymeric nanoparticles?

4.      Please define the magnetic nanoparticles and SPIONs cited in this work, MFe2O4 (M=Fe, Mn, Co, etc)?

5.      What is the effect of the particle size, shape, composition and concentration to be considered as nanocarrier and/or trigger? Please explain the trigger principle according to the nanoparticle material (pH, applied magnetic field AC, photo-dynamic interaction, ultrasound, etc.)

6.      Please indicate the particle size and shape for each nanoparticle: gold, silica, iron oxide, carbon, zinc oxide, etc.

7.      The challenges and perspectives should be included at the end of this review.

This work should be improved.

Author Response

Dear Authors,

The manuscript *Nanoparticles and mesemchymal stem cell (MSC) teraphy for cancer treatment: focus on nanocarriers and si-RNA chemokine CXCR4 blockade as strategies for tumor erradication in vitro an in vivo* is a good theme to be considered for a review. 

Some significant advances from Mesenchymal stem cells (MSCs) in combination with nanoparticles were shown to decrease the tumor growth in vitro and in vivo have been in this work. This manuscript should be improved to be considered in Micromachines, MDPI Journal.

  1. The manuscript has several typing errors along all the text. Please try to make all corrections from the title to the end of this work.

Dear reviewer

Firstable, thanks for all your comments, which help us to improve this R1 the content. Part of the content of our responses (size/shape…etc) have included in this R1 version (marked in red color).

The content has been revised  by an english mother native personne and typographic errors have been revised. Thanks again¡

  1. Almost all figures have the same features with errors in typewriting.

Typewriting errors haven been corrected and all corrected figures were pasted in this R1 version. Thanks again¡

  1. Covalent conjugation or physical association of nanoparticles to the surface of MSCs can significantly increase the drug load delivery by endocytosis. This is the main key of the discussion in this work.

We have added additional information in red color within this R1 version for its easy detection in the text. Please, does not forget the role of Nanoparticles (NP) and si-RNA or CXCR4 blocker as strategies for cancer treatment in this review.Thanks¡

All responses are indicated as follow:

Some bonding mechanisms between MSCs and Nanoparticles should be considered in the text like -OH,-COOH, etc.

These are several binding mechanisms between MSC and nanoparticles, which we have included within the entitled 3 point of this R1 version (¨MSC as vehicles for drug delivery against tumoral cells)¨ .For example, regarding stem cell differentiation, previous studies reported both positive and negative impacts of AgNPs on stem cell differentiation. As an example, AgNPs of size 10 or 20 nm and of size 30 nm are not toxic for MSCs cells and did not affect its differentiation.

Due to their unique characteristics, biocompatibility, and low toxicity, AuNPs have been regarded as favorable materials for directing stem cell fate and tissue regeneration. The shape, size, and surface characteristics of AuNPs impacted their potential to induce the osteogenic differentiation of MSCs. In human adipose-derived stem cells (hADSCs), the induction of osteogenic differentiation is induced by exposure to AuNPs (30 and 50 nm). Photo-curable gelatin hydrogels loaded with AuNPs markedly enhanced the proliferation and osteogenic differentiation of hADSCs. Several signalling pathways Wnt/β-catenin, ERK, p38) are involved in AuNP-induced osteogenic differentiation genes; in addition, L(D)-PAV-AuNPs-exposed MSCs showed upregulation of osteogenic differentiation marker genes; Another study reported the effect of a small size of AuNPs (4 nm) on the differentiation of hBM-MSCs compared with large size AuNPs (40 nm). The small AuNPs markedly suppressed osteogenic differentiation, while promoting the adipogenic differentiation of hBM-MSCs. Collectivelly, ROS mechanism is implicated in differentiation modulation in small size AuNP-treated hBM-MSCs. Moreover, AuNP-loaded functionalized nanofibrous scaffold promoted the cardiogenic differentiation of the MSCs, which was highlighted via morphological changes, formation of the contractile proteins, and the upregulation of the cardiogenic differentiation-related markers (all these cyted papers are included in the revisión by Ahmed Abdal Dayem and coworkers 2019. Efect of Metallic Nanoparticles on Stem Cell Proliferation and Differentiation 2019).

Ahmed Abdal Dayem, Soo Bin Lee and Ssang-Goo Cho. ¨The Impact of Metallic Nanoparticles on Stem Cell Proliferation and Differentiation¨. Nanomaterials 2018, 8, 761.

What is the binding mechanism affinity of MSCs with metallic nanoparticles and polymeric nanoparticles?

Interaction between nanoparticles (NP) and stem cells

The interaction of the nanomaterials with the cell membrane or intracellular components, as well as the ultimate modulation of a specific cellular signaling pathways by the internalized NPs (Illie et al 2012; Zhao et al., 2011). Clathrin and caveolin-dependent endocytosis, phagocytosis, macropinocytosis, and pinocytosis represent possible mechanisms for the cellular internalization of NPs (Bannunahet al., 2014, Yameen, et al., 2014; Oh et al., 2017; Wang, et al., 2019). Clathrin or caveolin-dependent endocytosis is considered to be the main mechanism for the uptake of nano-size materials (Wang et al, 2009, Pelkmans et al., 2001). The exocytosis or release of NPs is carried out via vesicle-dependent release, non-vesicle-dependent release, and lysosomal secretion. In addition, several inhibitors could modulate the mechanisms of cellular uptake, such as nocodazole, lovastatin, chlorpromazine, cytochalasin A, and genistein, which can be applied to characterize NP cellular internalization (Adjei, et al., 2014). The detailed mechanisms and pathways of nanomaterials exocytosis have been  descibed by Adjei, et al., 2014, and Ilie et a., 2012). The preparation of NPs, along with their physicochemical properties, is related to their biological function. Several mechanisms are involved in metallic NP-induced cellular proliferation and differentiation, oxidate reactive species and inflammatory mediators involved in the regulation of several transcription factors (Dolai et al., 2021; Ahmed Abdal Dayem, et al. 2018, mosto of cyted references in these R1 responses are included in the reference section of R1 submitted version, please see the bibliography of R1 version). The interaction of the nanomaterials with the cell membrane or intracellular components, as well as the activated cellular signaling pathways as a consequence of internalized NPs were reported in vitro (Illie et al 2012; Zhao et al., 2011). The preparation of NPs, along with their physicochemical properties, is related to their biological function. Several mechanisms are involved in metallic NP-induced cellular proliferation and differentiation, oxidate reactive species production as well as the regulation of various transcription factors (Ahmed Abdal Dayem et al., 2018) The binding mechanism affinity of MSCs with metallic nanoparticles and polymeric nanoparticles have been revised by Weiwei Wang et al., 2017). For example, silica-coated nagnetic Nanoparticles (NP) decrease human Bone Marrow-Derived Mesenchymal Stem Cell (hBMSC) Migratory Activity by reducing membrane fluidity and alters focal adhesion (Tae Hwan Shin, et al 2019). The silica-coated nagnetic NP decreased human hBMSC migratory capacity by reducing membrane fluidity and impairs focal adhesion molecules (Ahmed Abdal Dayem et al., 2018). Regarding stem cell differentiation, previous studies have demonstrated positive and negative impacts of AgNPs on stem cell differentiation. As an example, AgNPs of size 10 or 20 and 30 nm are not toxic for MSCs cells and did not affect its differentiation. The shape, size, and surface characteristics of AuNPs impacted their potential to induce the osteogenic differentiation of MSCs. In human adipose-derived stem cells (hADSCs), the induction of osteogenic differentiation is prompted by AuNPs (sized 30 and 50 nm). Photo-curable gelatin hydrogels loaded with AuNPs increased the proliferation and osteogenic differentiation of hAD-MSCs. In human BM-MSC (hBM-MSCs) and MC3T3-E1 cells, miR029b-delivered polyethyleneimine (PEI)-capped AuNPs efficiently promoted the osteogenic differentiation without signifficant toxicity; In addition, L(D)-PAV-AuNPs-exposed MSCs showed upregulation of osteogenic differentiation marker genes; another study the small size of AuNPs (4 nm) on the differentiation of hBM-MSCs compared with large size AuNPs (40 nm). The small AuNPs markedly suppressed osteogenic differentiation and promoted the adipogenic differentiation of hBM-MSCs. Collectivelly, the differentiation of stem cells in small size AuNP-treated hBM-MSC is influenced by oxidative stress and inflammatory mediators. Interestingly, AuNP-loaded functionalized nanofibrous scaffold promoted the cardiogenic differentiation of the MSCs by upregulating cardiogenic differentiation-related markers (all these studies were included by Ahmed Abdal Dayem and coworkers 2019). The binding mechanism affinity of MSCs with metallic nanoparticles and polymeric nanoparticles has been studied by Weiwei Wang and coworkers 2007). In adidition, The treatment with several inhibitors (nocodazole, lovastatin, chlorpromazine, cytochalasin A, and genista) could affect the cellular uptake in vitro and are useful for study the NP cellular internalization (Oh et al., 2019; Pelkmans et al., 2001; Wang et al., 2009; Yameen, B et al., 2014; Zhao et al., 2011).

It is known that conjugation of polyethylene glycol to the surface of NPs is one of the key tools for surface modifications of the carriers. As an example, the AuNP represents a model metallic NP with high efficiency for delivering nucleic acids, recombinant proteins, and drug compounds to the tumor (Kong et al., 2017). As an anti-cancer strategy, the delivery of methotrexate, a well-known anti-cancer drug, was conducted via the binding of its carboxyl group with the surface of AuNP (Chen et al., 2007). For additional details on the possible application of noble metallic NPs in drug delivery for cancer therapy consult Wang et al., 2011, Ahmad et al., 2010. These treatments with AgNPs of size 10, 20 and 30 nm are not toxic for MSCs and also increase the adipogenic differentiation of hBM-MSCs of urine-derived stem cells (Illie et al., 2012; Oh et al., 2019; Pelkmans et al., 2001; Wang et al., 2009; Yameen, B et al., 2014; Zhao et al., 2011; Kong et al., 2017; Dolai et al., 2021). Cell membrane-coated nanoparticles are emerging as a new type of promising nanomaterials for targeted delivery and could also evade the immune system response. In a study, two mesenchymal stem cell (MSC) membrane-coated silica nanoparticles (MCSNs) were synthesized, which have similar sizes but different stiffness values (MPa and GPa). Unexpectedly, a much lower macrophage uptake, but much higher cancer cell uptake, was found with the soft MCSNs compared with the stiff MCSNs. Intriguingly, the soft MCSNs increased the formation of a more protein-rich membrane coating rich in CXCR4 and MSC surface marker CD90. This led to the soft MCSNs enhancing cancer cell uptake mediated by CXCR4/SDF-1 signalling pathways. These findings add information on the combination of nanoparticle elasticity and promote bio-nano interactions for more effective nanomedicines against tumor progression. These strategies should be studied in more detail by analyzing a better homing of MSC in the tissue and the  possible long-term toxicity should not be excluded (Da Zou et al., 2023; these cyted references are included in this R1 version).

Adjei, I.M.; et al. Nanoparticles: Cellular uptake and cytotoxicity. In Nanomaterial; Springer: Berlin, Germany, 2014; pp. 73–91

Ahmad, M.Z.; et al. Metallic nanoparticles: Technology overview & drug delivery applications in oncology. Expert Opin. Drug Deliv. 2010, 7, 927–942.

Ahmed Abdal Dayem, et al. The Impact of Metallic Nanoparticles on Stem Cell Proliferation and Differentiation. Nanomaterials (Basel). 2018 Oct; 8(10): 761.

Bannunah, A.M.; et al. Mechanisms of nanoparticle internalization and transport across an intestinal epithelial cell model: Effect of size and surface charge. Mol. Pharmaceutics 2014, 11, 4363–4373. 

Caruso, D.M.; et al. Randomized clinical study of hydrofiber dressing with silver or silver sulfadiazine in the management of partial-thickness burns. J. Burn Care Res. 2006, 27, 298–309

Chen, Y.-H.; et al. Methotrexate conjugated to gold nanoparticles inhibits tumor growth in a syngeneic lung tumor model. Mol. Pharmaceutics 2007, 4, 713–722.

Da Zou, Zeming Wu et al. Dual functions of silver nanoparticles in f9 teratocarcinoma stem cells, a suitable model for evaluating cytotoxicity-and differentiation-mediated cancer therapy. Int. J. Nanomed. 2017, 12, 7529

He, W.; et al. Silver nanoparticle based coatings enhance adipogenesis compared to osteogenesis in human mesenchymal stem cells through oxidative stress. J. Mater. Chem. B 2016, 4, 1466–1479.

Ilie, I.; et al. Influence of nanomaterials on stem cell differentiation: Designing an appropriate nanobiointerface. Int. J. Nanomed. 2012, 7, 2211.

It is known that conjugation of polyethylene glycol to the surface of NPs was one of the key tools for surface modifications of the carriers. As an example, the AuNP represents a model metallic NP, showing high efficiency for delivering recombinant proteins, nucleic acids, and drug compounds to a target area while controlling the release of the delivered compounds (Kong et al., 2017). As an anti-cancer strategy, the delivery of methotrexate, a well-known anti-cancer drug, was conducted via the binding of its carboxyl group with the surface of AuNP (Chen et al., 2007). Additional details on the possible application of noble metallic NPs in drug delivery for cancer therapy was described in some studies (Wang et al., 2011; Ahmad et al., 2010)

Kong, F.-Y.; et al. Unique roles of gold nanoparticles in drug delivery, targeting and imaging applications. Molecules 2017, 22, 1445.

Liu, X.; et al. Influence of silver nanoparticles on osteogenic differentiation of human mesenchymal stem cells. J. Biomed. Nanotechnol. 2014, 10, 1277–1285.

Oh, N.; Park, J.-H. Endocytosis and exocytosis of nanoparticles in mammalian cells. Int. J. Nanomed. 2014, 9, 51.

Pelkmans, L.; et al. Caveolar endocytosis of simian virus 40 reveals a new two-step vesicular-transport pathway to the er. Nat. Cell Biol. 2001, 3, 473

Qin, H.; et al. Silver nanoparticles promote osteogenic differentiation of human urine-derived stem cells at noncytotoxic concentrations. Int. J. Nanomed. 2014, 9, 2469.

Qureshi, A.T.; et al. Mir-148b–nanoparticle conjugates for light mediated osteogenesis of human adipose stromal/stem cells. Biomaterials 2013, 34, 7799–7810. 

Tae Hwan Shin, et al. Silica-Coated Magnetic Nanoparticles Decrease Human Bone Marrow-Derived Mesenchymal Stem Cell Migratory Activity by Reducing Membrane Fluidity and Impairing Focal Adhesion. Nanomaterials (Basel) 2019 Oct 17;9(10):1475.

Wang, F.; et al. Doxorubicin-tethered responsive gold nanoparticles facilitate intracellular drug delivery for overcoming multidrug resistance in cancer cells. ACS Nano 2011, 5, 3679–3692.

Wang, Z.; et al. Size and dynamics of caveolae studied using nanoparticles in living endothelial cells. ACS Nano 2009, 3, 4110–4116. 

Weiwei Wang, et al. Functional Nanoparticles and their Interactions with Mesenchymal Stem Cells. Curr Pharm Des. 2017;23(26):3814-3832.

Xia Zhao. Nanoparticle elasticity regulates the formation of cell membrane-coated nanoparticles and their nano-bio interactions. Proc Natl Acad Sci U S A 2023 Jan 3;120(1):e2214757120.

Yameen, B.; et al. Insight into nanoparticle cellular uptake and intracellular targeting. J. Control. Release 2014, 190, 485–499.

Zhao, F.; et al. Cellular uptake, intracellular trafficking, and cytotoxicity of nanomaterials. Small 2011, 7, 1322–1337. 

Most of these cyted references are included in the review 74 by Sousa de Almeida M, et al . Understanding nanoparticle endocytosis to improve targeting strategies in nanomedicine. Chem Soc Rev. 2021 May 7;50(9):5397-5434) and were also included in this R1 version.

Humans are increasingly exposed to nanoparticles (NPs) that originate from various manufactured products, such as pesticides, food, textiles, cosmetics, or paints. This growing exposure to NPs calls into question their impact on human health. Metallic nanoparticles are present in the environment, raising their potential toxicity on live organisms. Certain metals (i.e., copper, zinc, silver, cadmium, and gold) trigger a common cell response metallothioneins dependent levels (Alice Balfourier et al 2022. Importance of Metal Biotransformation in Cell Response to Metallic Nanoparticles: A Transcriptomic Meta-analysis Study. ACS Nanosci Au 2022 Sep 30;3(1):46-57). The cellular response to metallic compounds is complex to capture, as it may vary not only with the nature of the metal, but also with its chemical formulation (coordination sphere, counterions, oxidation state, crystallization, size, shape or surface state of NPs. This complexity makes it difficult to identify common patterns in the cell response to metallic compounds (Alice Balfourier et al., 2023).

Among these NPs, metal-based NPs (i.e., metallic, metal oxide, or metal sulfide NPs) represent an important proportion, but their nanotoxicology is still a concern. Metallic pollutants could be toxic for humans. The NCBI (National Center for Biotechnology Information) database GEO (Gene Expression Omnibus) allow the identification ¨in vitro¨ data sets that describe human cell response to single exposure to a metallic compound (metallic NPs or metal ions). To investigate nonmetallic NPs as control, the authors have chossen 5 additional data sets of cells exposed to carbon-based nanomaterials or silicon-based NPs. The selected data sets most often compared multiple doses, time points after exposure, cell types, and NPs with different sizes, coatings, and shapes. The cell transcriptomic response was evaluated and compared with different sizes, shapes, and coating of NPs constituted of the same material by Pearson’s correlation. Regarding metal ions, differences could also be expected, for example, in the case of Pt between tumor cell nonmetallic NPs (Alice Balfourier et al., 2022). These authors analyzed the posible correlation between cell responses to nanoparticulate or ionic forms of the same metal by r Pearson correlation. Remarkably, the correlation coefficient between NPs and ions exceeds 0.6 for Ag, Cu, and Zn (rAg(ion)/Ag(NP) = 0.83, rCu(ion)/Cu(NP) = 0.66, rZn(ion)/Zn(NP) = 0.64) and 0.4 for Fe and Au (rFe(ion)/Fe(NP) = 0.48, rAu(ion)/Au(NP) = 0.47) (for review consult Alice Balfourier et al., 2023).

Alice Balfourier , et al. Importance of Metal Biotransformation in Cell Response to Metallic Nanoparticles: A Transcriptomic Meta-analysis Study. ACS Nanosci Au 2022 Sep 30;3(1):46-57. 

Almeida Mauro Sousa de, Eva Susnik, Barbara Drasler, Patricia Taladriz-Blanco, Alke Petri-Fink and Barbara Rothen-Rutishauser. Understanding nanoparticle endocytosis to improve targeting strategies in nanomedicine. Chem Soc Rev. 2021 May 11; 50(9): 5397–5434.

Weiwei Wang, Zijun Deng, Xun Xu, Zhengdong Li, Friedrich Jung, Nan Ma, Andreas Lendlein. Functional Nanoparticles and their Interactions with Mesenchymal Stem Cells. Curr Pharm Des. 2017;23(26):3814-3832.

  1. Please define the magnetic nanoparticles and SPIONs cited in this work, MFe2O4 (M=Fe, Mn, Co, etc)?

SPIO Superparamagnetic iron oxide (SPIO NPs) are a type of iron oxide NPs (IONPs) that possess superparamagnetism property (LaConte, et al., 2005; Bull, E.; et al., 2014) when are subjected to an external magnetic field. For example, coating of SPIO with dextran (DEX) and then labeling of stem cells led to a marked induction of myogenic differentiation under pulsed electromagnetic field, which was evidenced through the upregulation of the expression level of the myogenic-specific markers (MyoG and Myh2) (Norizadeh-Abbariki, T.; et al., 2014). Interestingly, chemo-attractants, namely, chemokine receptor type 4 (CXCR4) and epidermal growth factor receptor (EGFR) can recruit DEX-IONP-labeled hMSC to the site of damage (Chung, T.-H et al., 2018). The magnetic field-induced assembly (stripe-like) of magnetic IONPs was exploited for the conversion of primary mouse bone marrow cells into osteoblasts (Sun, J.; et al., 2014). SPIO NPs are considered as being a safe nanomaterial for stem cell labelling. The interface between IONP magnetic assemblies and the cells is implicated in the induction of osteogenic differentiation rather than the particles internalization into the the human bone marrow stem cells (BM-MSCs). Interestingly, the magnetic field-induced assembly (stripe-like) of magnetic IONPs was exploited for the conversion of primary mouse bone marrow cells into osteoblasts (Sun, J.; et al., 2014). The interface between IONP magnetic assemblies and the stem cells ontribute to the induction of osteogenic differentiation rather than the particles internalization into the hBM-MSCs cells (Wang, Q.; et al., 2016); In addition, IONP-induced gap junction communication between cardiomyoblasts and MSCs lead to therapeutic effects in a rodent model of myocardial infarction (Han, J et al., 2015) while SPIO NPs induced the proliferation of hMSCs by regulating cell cycle-related proteins under oxidative stress (Huang, D.-M et al., 2019).

To date, the uptake of different NPs through clathrin/caveolae-independent endocytosis has been reported. SPIONs and silica-coated iron oxide NPs (Fe3O4@SiO2) with negative surface charge and a primary diameter of around 17 to 30 nm were shown to be internalized via CDC42 (CLIC–GEEC pathway) and caveolae in HeLa cells (Le Conte et al., 2005). On the other hand, co-exposure with different nanoparticles. Rafieepour et al. investigated the in vitro toxicological effects of single and combined exposure of magnetite (Fe3O4) NPs and polymorphous silica NPs on human epithelial cell line A549.367. The cells were exposed to four different NP concentrations (10, 50, 100, and 250 mg mL1) of both NP types simultaneously for 24 h and 72 h. They found toxic effects after exposuring Fe3O4 and SiO2 NPs but the combined exposure to Fe3O4 and SiO2 NPs led to antagonistic interactions by reducing toxicity in comparison to single exposures. This effect can be explained by the accumulation of intracellular proteins on the SiO2 surface, forming a protein corona. Fe3O4 could provoke the synthesis of cellular proteins, causing the formation of protein corona on the surface of silica NPs and thereby reducing its cytotoxic effect (Okoturo-Evans et al., 2013; Lesniak A et al., 2012). The formation of a corona can reduce cellular uptake of functionalized NPs by shielding the ligands from binding to their receptors. As an example, the attachment of serum proteins on the surface of transferrin-functionalized SiO2 NPs resulted in a loss of its targeting capability (Salvati A, et al., 2013). When comparing in vivo and in vitro conditions, differences in NPs uptake should be considered due to variations in serum protein concentrations (in vivo, NPs in blood encounter much higher serum concentrations). To date, the uptake of different NPs through clathrin/caveolae-independent endocytosis has been reported. SPIONs and silica-coated iron oxide NPs (Fe3O4@SiO2) with negative surface charge and a primary diameter of around 17 to 30 nm were shown to be internalized via CDC42 (CLIC–GEEC pathway) and caveolae in HeLa cells (Le Conte et al., 2005; Almeida et al., 2020).

Lesniak A., et al. ACS Nano, 2012, 6, 5845–5857.

Okoturo-Evans O., et al. PLoS One, 2013, 8, e72363.

Salvati A, et al. Transferrin-functionalized nanoparticles lose their targeting capabilities when a biomolecule corona adsorbs on the surface. Nat Nanotechnol. 2013 Feb;8(2):137-43

  1. What is the effect of the particle size, shape, composition and concentration to be considered as nanocarrier and/or trigger?

The International Organization for Standardization (ISO) defines a nanomaterial as a “material with any external dimension in the nanoscale or having an internal structure or surface structure in the nanoscale (1–100 nm)”. Similarly, in 2011, the European Commission adopted a definition for a nanomaterial: “A natural, incidental or manufactured material containing particles, in an unbound state or as an aggregate or as an agglomerate and where, for 50% or more of the particles in the number size distribution, one or more external dimensions is in the size range 1–100 nm”. The United States Food and Drug administration (US FDA) states that nanomaterials are “materials up to one micron if these ones exhibit properties or phenomena that are attributable to its dimensions”. In this review, a size range between 1–1000 nm is considered for nanomaterials and nanoparticles (NPs). Mauro Sousa de Almeida et al considered, a size range between 1–1000 nm for nanomaterials and nanoparticles (NPs) (de Almeida et al., 2020).

Boverhof D. R et al. Regul. Toxicol. Pharmacol., 2015, 73, 137 —150

Almeida, Mauro Sousa et al considered, a size range between 1–1000 nm for nanomaterials and nanoparticles (NPs) (Almedida et al., 2020). For example, phagocytosis encompasses the uptake of large particles (> or equal to 0.5 mm) and is only performed by specialized cells.

EC, Commission recommendation of 18 October 2011 on the definition of nanomaterial (2011/696/EU), 2011.

Nanotechnologies,Vocab (Part 1 Core terms), 2015.

These are possible entry mechanisms for nanoparticles (NPs). Large NPs (4500 nm) and aggregates enter the cell through phagocytosis (0.11 microm) and macropinocytosis (0.2-1 microm). NP opsonization via IgG leads to cellular recognition (FcgR) in phagocytes.  Pinocytosis includes different mechanisms: macropinocytosis, clathrin-mediated endocytosis (100-150 nm), caveolae-dependent endocytosis (60-90 nm), CLIC–GEEC (40-80 nm), flotillin-assisted endocytosis (< 100 nm), fast endophilin-mediated endocytosis (< 1 microm), RhoA-dependent endocytosis (<200 nm) and Arf-6-associated endocytosis (< 100 nm). As a non-selective endocytic process, macropinocytosis (0.2-5 microm) is associated with the internalization of different NPs. Smaller NPs (10 nm) and cationic NPs, with high charge density, enter the cell via direct penetration and pore formation , respectively. NPs functionalized with transferrin and albumin are taken up through clathrin-mediated and caveolae-dependent endocytosis, respectively (Mauro Sousa de Almeida et al., 2021).

The protein composition and conformation on the NPs surface influence the interaction and recognition by the phagocyte surface receptors. The receptors involved in this recognition dictate the subsequent signaling cascade and may potentially initiate inflammatory events (e.g., FcR).As an example, Silica (SiO2) NPs of 50 and 100 nm triggered inflammation in THP-1 macrophage-like cells by the activation of the scavenger receptor A1. In a different context, a pre-coating of SiO2 NPs and single-carbon nanotubes coated with the surfactant Pluronic F127 reduced the adsorption of serum proteins and inhibited the anti-inflammatory effect on murine macrophages (RAW 264.7 cells).

.The route of internalization of 40 nm polystyrene (PS) NPs was studied in different cell types. Caveolae-dependent endocytosis is a common route of internalization of NPs.Caveola is the designation for the flask-shaped invaginations with 50–100 nm that can be found in the plasma membrane of specific mammalian cells. The CLIC–GEEC pathway is a dynamin-independent process leading to the formation of tubular/ring-like invaginations of the plasma membrane of around 200–600 nm in length and 40–80 nm in width. Similarly, Arf6-associated endocytosis together with caveolae-dependent endocytosis and macropinocytosis are associated with the uptake of 130 nm polydopamine-coated mesoporous silica NP; the passive uptake was observed in lung cells after exposure of titanium dioxide (TiO2) NPs of 22 nm to rats via inhalation (for review consult Almeda el al., 2021). The lower uptake could be related to differences in the corona composition (for revision consult Sousa de Almeida M, et al. 2021 May 7;50(9):5397-5434).

The effect of Nanoparticle concentration and signaling pathways are variable depending of type of NP and stem cell considered (for revision consult Ahmed Abdal Dayem et al., 2018). For example, AuNP-treated induce osteogenic differentiation of stem cells (Xiang et al., 2018). The shape, size, and surface characteristics of AuNPs affect the osteogenic differentiation of MSCs; Rod-shaped AuNP with a size of 70 nm markedly promoted the osteogenic differentiation, while 40 nm rod-shaped AuNPs suppressed osteogenic differentiation (Li et al,. 2016). In human adipose-derived stem cells (hADSCs), the induction of osteogenic differentiation is prompted upon exposure to AuNPs sized 30 and 50 nm (Ko et al., 2012). In human BM-MSC (hBM-MSCs) and MC3T3-E1 cells, miR029b-delivered polyethy leneimine (PEI)-capped AuNPs efficiently promoted the osteogenic differentiation without toxicity; the osteogenic differentiation-inducing capacity was mediated by increases of the osteogenic differentiation-related genes, namely alkaline phosphatase (ALP), osteopontin (OPN), osteocalcin (OCN), and Runt-related transcription factor 2 (RUNX2) (Pan et al., 2016). Another study evaluated the small size of AuNPs (4 nm) on the differentiation of hBM-MSCs compared with large size AuNPs (40 nm). The small AuNPs markedly suppressed osteogenic differentiation, while promoting the adipogenic differentiation of hBM-MSCs (Dayem et al., 2016).

Dayem, A.A.; et al. The potential of nanoparticles in stem cell differentiation and further therapeutic applications. Biotechnol. J. 2016, 11, 1550–1560

Ko, W.et al The effect of gold nanoparticle size on osteogenic differentiation of adipose-derived stem cells. J. Colloid Interface Sci. 2015, 438, 68–76

Li, J.; et al. Gold nanoparticle size and shape influence on osteogenesis of mesenchymal stem cells. Nanoscale 2016, 8, 7992–8007

Pan, T.; et al. Mir-29b-loaded gold nanoparticles targeting to the endoplasmic reticulum for synergistic promotion of osteogenic differentiation. ACS Appl. Mater. Interfaces 2016, 8, 19217–19227

Xiang, Z.; et al. Gold nanoparticles inducing osteogenic differentiation of stem cells: A review. J. Cluster Sci. 2018, 29, 1–7

TiO2

The internalization rate of polystyrene NPs and polystyrene NPs functionalized with an amine group in MSCs has been studied in vitro (Van, V.T.; et al., 2007). Amino-functionalized polystyrene NPs showed faster internalization and higher cellular uptake in comparison to unfunctionalized polystyrene NPs. For example, TiO2 NPs are toxic for MSCs in a size-dependent manner by reducing cell migration, promoting a lack of cell membrane integrity, and also suppress the osteogenic differentiation (Hou, Y.; et al., 2013). TiO2 nanotubes larger than 50 nm showed a drastically decrease in the proliferation and differentiation of MSCs (Park, J., et al., 2007). In addition, TiO2–COOH nanorods impeded the osteogenic differentiation of rBM-MSCs (Shrestha et al., 2016, for review consult Almeida., 2020).

Hou, Y.; et al. Effects of titanium nanoparticles on adhesion, migration, proliferation, and differentiation of mesenchymal stem cells. Int. J. Nanomed. 2013, 8, 3619.

Park, J.; et al. Nanosize and vitality: TiO2 nanotube diameter directs cell fate. Nano Lett. 2007, 7, 1686–1691.

Shrestha, S.; et al. Citrate-capped iron oxide nanoparticles impair the osteogenic differentiation potential of rat mesenchymal stem cells. J. Mater. Chem. B 2016, 4, 245–256.

Tekin, H.; et al. Controlling spatial organization of multiple cell types in defined 3d geometries. Adv. Mater. 2012, 24, 5543–5547.

Zorlutuna, P.; et al. Microfabricated biomaterials for engineering 3d tissues. Adv. Mater. 2012, 24, 1782–1804.

TiO2 NPs induces the neuronal differentiation of the mNSCs (Liu et al., 2020). In additon, small-sized TiO2 nanotubes (15–30 nm) underpinned cell adhesion and spreading and therefore promoted the osteogenic differentiation of rat BM-MSCs (rBM-MSCs) (Park et al., 2007). Titanium nanotubes promoted the osteogenesis of pulp and adipose tissue-derived stem cells (Pozio, A et al., 2012). TiO2–NH2, TiO2–COOH, and TiO2–PEG did not show any negative effects on the adipogenic differentiation of rBM-MSCs (Shrestha et al., 2016). The promotion of osteogenic differentiation of hBM-MSCs cultured on TiO2 surface was mediated the high level of phosphorylation of focal adhesion kinase (FAK). TiO2 nanotube of size 70 nm are ideal for induction of osteogenic differentiation of hASCs (Adipose stem cells (Lv et al., 2015). As a whole, metallic NPs can exploit epigenetic mechanisms for the modulation of stem cell differentiation. As an example, TiO2 nanotube-induced osteogenic differentiation of bone marrow stromal cells (Yu, W.; et al., 2015).

Liu, X.; et al. A protein interaction network for the analysis of the neuronal differentiation of neural stem cells in response to titanium dioxide nanoparticles. Biomaterials 2010, 31, 3063–3070.

Lv, L.; et al. The nanoscale geometry of TiO2 nanotubes influences the osteogenic differentiation of human adipose-derived stem cells by modulating h3k4 trimethylation. Biomaterials 2015, 39, 193–205

Park, J.; et al. Nanosize and vitality: TiO2 nanotube diameter directs cell fate. Nano Lett. 2007, 7, 1686–1691

Pozio, A.; et al. Titanium nanotubes stimulate osteoblast differentiation of stem cells from pulp and adipose tissue. Dent. Res. J. 2012, 9, S169–S174

Shrestha, S.; et al. Influence of titanium dioxide nanorods with different surface chemistry on the differentiation of rat bone marrow mesenchymal stem cells. J. Mater. Chem. B 2016, 4, 6955–6966

Yu, W.; et al. Mechanisms of stem cell osteogenic differentiation on TiO2 nanotubes. Colloids Surf. B 2015, 136, 779–785.

IONPs

The intracerebral (DEX-IONP-hMSCs) injection into the right ventricle lead to neurorepair effect when tiroxine hidrolase was evaluated in a rodent model of Parkinson disease by 6-Hidroxidopamine infusion (Chung, T.-H.; et al., 2018). On the other hand, IONP-grown hBM-MSCs showed enhanced osteogenic differentiation by MAPK signal pathways in a dose-dependent manner. In this study., the magnetic field-induced assembly (stripe-like) of magnetic IONPs was exploited for the conversion of primary mouse bone marrow cells into osteoblasts (Sun et al., 2014). The interface between IONP magnetic assemblies and the cells is implicated in the induction of osteogenic differentiation rather than the particles internalization into the cells. Magnetic IONPs were shown to boost the expression level of the long noncoding RNA INZEB2, which ultimately promoted the osteogenic differentiation of hBM-MSCs (Wang Q et al., 2017). In addition, IONPs promoted potential crosstalk between the cardiomyoblasts and MSCs via the upregulation of the expression of a gap junction protein, namely connexin 43 in IONP-exposed cardiomyoblasts (all these references have been included within the review by Almeida et a., 2020).

Chung, T.-H.; Hsu, S.-C.; Wu, S.-H.; Hsiao, J.-K.; Lin, C.-P.; Yao, M.; Huang, D.-M. Dextran-coated iron oxide nanoparticle-improved therapeutic effects of human mesenchymal stem cells in a mouse model of parkinson’s disease. Nanoscale 2018, 10, 2998–3007.

Sun, J.; et al. Magnetic assembly-mediated enhancement of differentiation of mouse bone marrow cells cultured on magnetic colloidal assemblies. Sci. Rep. 2014, 4, 5125.

Wang, Q.; Chen, B.; Ma, F.; Lin, S.; Cao, M.; Li, Y.; Gu, N. Magnetic iron oxide nanoparticles accelerate osteogenic differentiation of mesenchymal stem cells via modulation of long noncoding rna inzeb2. Nano Res. 2017, 10, 626–642

 Other Metallic NPs

 The differentiation of hMSCs into osteoblasts was enhanced upon their culture with polymeric fibrous polyethersulfone-polyethylene glycol (PES-PEG) electrospun composites coated with willemite or Zn2SiO4 bioceramic NPs by induding the expression osteogenesis-related markers (Amiri et al., 2016). The topographical orientation of ZnO NPs on the surface had a significant impact on stem cell differentiation by increasing the neural survival (Amiri et al., 2016; Ansari et al., 2016). In addition SiNP of 50–120 nm stimulated the proliferation of hADSCs as well as promoted the osteogenesis of hMSCs (Yang, X et al. ,2016; Kim et al., 2015). Moreover, SiNP-treated hMSCs showed high focal adhesion and upregulated expression he connexin-43 as regenerative mechanisms; interestingly, the conjugation of SiNP with insulin markedly enhanced the adipogenic differentiation of rMSCs with minimal cytotoxicity (Popara et al., 2018; Liu et al., 2010).

Amiri, B.; et al. Osteoblast differentiation of mesenchymal stem cells on modified pes-peg electrospun fibrous composites loaded with Zn2SiO4 bioceramic nanoparticles. Differentiation 2016, 92, 148–158.

Ansari, F.; Njuguna, J.; Kavosh, N.; Briscoe, J. Correlation between stem cell differentiation and the topography of zinc oxide nanorods. J. Bionanosci. 2015, 9, 73–76

Kim, K.J.; et al. Silica nanoparticles increase human adipose tissue-derived stem cell proliferation through erk1/2 activation. Int. J. Nanomed. 2015, 10, 2261

Liu, D.; et al. Biocompatible silica nanoparticles− insulin conjugates for mesenchymal stem cell adipogenic differentiation. Bioconjugate Chem. 2010, 21, 1673–1684

Popara, J.; et al. Silica nanoparticles actively engage with mesenchymal stem cells in improving acute functional cardiac integration. Nanomedicine 2018, 13, 10.

Yang, X.; et al. The stimulatory effect of silica nanoparticles on osteogenic differentiation of human mesenchymal stem cells. Biomed. Mater. 2016, 12, 015001.

Metallic NPs and Stem Cell Toxicity

Metallic NPs induced positive effects on stem cell differentiation although also reported some adverse effects since AgNP-induced neurotoxicity by oxidative stress, mitochondrial dysfunction, and regulation of apoptosis (Liu et al., 2015). The toxicity of 30 nm AgNPs and its role in adipogenic differentiation of hBM-MSCs has been reported (He et al., 2016). AgNPs enhanced toxicity but cells exposed to 25 and 50 µg/mL AgNPs for 24 h did not show marked toxicity. Co-treatment with ascorbic acid (antioxidant), abolished the neurotoxic action of AgNPs while AgNPs toxicity of size 80 nm on the osteogenic and adipogenic differentiation of hMSCs was previously reported (Sengstock et al., 2014).

Liu, F.; et al. Effects of silver nanoparticles on human and rat embryonic neural stem cells. Front. Neurosci. 2015, 9, 115

He, W.; et al. In vitro effect of 30 nm silver nanoparticles on adipogenic differentiation of human mesenchymal stem cells. J. Biomed. Nanotechnol. 2016, 12, 525–535.

Sengstock, C.; et al. Effect of silver nanoparticles on human mesenchymal stem cell differentiation. Beilstein J. Nanotechnol. 2014, 5, 2058

SPIO NPs

The studies on the toxicity of IONPs or SPIO NPs in stem cell need further investigation;  the special physico-chemical properties of SPIO NPs, such as large surface area and the enhanced reactivity of its surface provoques cytotoxicity (Singh et al., 2010). The surface chemistry alteration of IONP via capping with citrate significantly hampered osteogenic differentiation of MSCs by reducing the expresión of osteogenic differentiation-related genes; in contrast, IONPs coated with pristine showed no significant suppression of osteogenic differentiation of rMSCs (Singh, et al., 2010). SPIO NPs (Ferucarbotran) showed a concentration-dependent suppression to the osteogenic differentiation of hMSCs (Chen et al., 2010). High concentration of SPIO NPs (300 µg/mL) abrogated the osteogenic differentiation and enhanced the cell migration; finally, free iron is involved in SPIO NP-induced inhibition of the osteogenic differentiation of hMSCs. On the other hand, SPIO (Feridex)-labeled hBM-MSCs showed no alteration in the cell proliferation, and osteogenic or adipogenic differentiations, but supressed the chondrogenic differentiation (Kostura et al., 2004).

Chen, Y.-C.; et al. The inhibitory effect of superparamagnetic iron oxide nanoparticle (ferucarbotran) on osteogenic differentiation and its signaling mechanism in human mesenchymal stem cells. Toxicol. Appl. Pharmacol. 2010, 245, 272–279.

Kostura, L.; et al. Feridex labeling of mesenchymal stem cells inhibits chondrogenesis but not adipogenesis or osteogenesis. NMR Biomed. 2004, 17, 513–517

Shrestha, S.; et al. Citrate-capped iron oxide nanoparticles impair the osteogenic differentiation potential of rat mesenchymal stem cells. J. Mater. Chem. B 2016, 4, 245–256.

Singh, N.; et al. Potential toxicity of superparamagnetic iron oxide nanoparticles (spion). Nano Rev. 2010, 1, 5358.

Syama, S.; et al. Zinc oxide nanoparticles induced oxidative stress in mouse bone marrow mesenchymal stem cells. Toxicol. Mech. Methods 2014, 24, 644–653.

Shape

The size of nanocarrier could be 30–50 nm or 50–200 nm. Therefore, particle size and shape influence the biological function and toxicity of NPs and these factors should be considered during the design of nanomaterials; i The size, shape, surface properties, stiffness, and the hydrophilic or hydrophobic properties of NPs are important for internalization into the cells. In fact, the size, shape, surface properties, stiffness, and the hydrophilic or hydrophobic properties of NPs are important for internalization into the cells (Kerativitayanan, P.; et al., 2015).For example, the uptake of NPs is inversely correlated with the particle size. In this context, higher uptake of smaller size NPs (30–50 nm) was reported, compared to bigger size NPs (50–200 nm), which show less cellular internalization (Li J et al., 2016, Ko et al., 2015). Spherical-shaped NPs showed higher uptake rate than that that of non-spherical NPs [Florez, L. et al., 2012, Zhang, S  et al., 2015). The chemical modification of NPs by increasing the softness and the hydrophobicity has been shown to lead to a high rate of internalization (Lorenz, S et al., 2010). Of note, a particle surface charge is implicated with cellular internalization rates. NPs possessing a positive charge can quickly enter the nucleus and avoid the lysosomal degradation, whereas particles with negative or neutral charges can easily localize to the lysosome instead of at the perinuclear región (Zhang, R et al., 2015). Preparation of engineered NPs with the desired functional group is one of the recent tools for modulation of cellular events for a particular biomedical application (Tekin, H.; et al., 2006, Zorlutuna, P et al, 2007). In this regard, a research group compared the internalization rate of polystyrene NPs and polystyrene NPs functionalized with an amine group in MSCs (Van, V.T.; et al., 2007). Amino-functionalized polystyrene NPs showed faster internalization and higher cellular uptake than that of unfunctionalized polystyrene NPs. Clathrin-dependent endocytosis was the main mechanism in the internalization of the amino-functionalized polystyrene NPs (Van, V.T.; et al., 2007).

For example, the negative influence of TiO2 NPs on the proliferation of MSCs was also reported. TiO2 NPs showed a toxic effect in MSCs in a size-dependent manner, which was highlighted by low cell migration, lack of cell membrane integrity, and suppression of the osteogenic differentiation (Hou, Y.; et al., 2013). TiO2 nanotubes larger than 50 nm showed a drastic decrease in the proliferation and differentiation of MSCs (Park, J., et al., 2007). TiO2–COOH nanorods impeded the osteogenic differentiation of rBM-MSCs that contributed to the induction of the expression level of fibroblast growth factor (FGF-2) and transforming growth factor beta 1 (TGF-β1) (Shrestha et al., 2016).

It is noteworthy that ROS modulation represents one of the key mechanisms of metallic NP-associated cellular functions (Abdal Dayem, A et al., 2020). In addition, ROS generation is implicated in the modulation of stem cell differentiation (Abdal Dayem, A 2020, Mody, et al., 2001). Collectively, particles with tailored physicochemical properties can exert differential influence on stem cell differentiation and proliferation.

Abdal Dayem, A.; et al. The role of reactive oxygen species (ros) in the biological activities of metallic nanoparticles. Int. J. Mol. Sci. 2017, 18, 120.

Florez, L.; et al. How shape influences uptake: Interactions of anisotropic polymer nanoparticles and human mesenchymal stem cells. Small 2012, 8, 2222–2230.

Hou, Y.; et al. Effects of titanium nanoparticles on adhesion, migration, proliferation, and differentiation of mesenchymal stem cells. Int. J. Nanomed. 2013, 8, 3619.

Jiang, P.; et al. Fe3O4/bsa particles induce osteogenic differentiation of mesenchymal stem cells under static magnetic field. Acta Biomater. 2016, 46, 141–150. 

Kerativitayanan, P.; et al. Nanomaterials for engineering stem cell responses. Adv. Healthc. Mater. 2015, 4, 1600–1627

Ko, W.-K.; et al. The effect of gold nanoparticle size on osteogenic differentiation of adipose-derived stem cells. J. Colloid Interface Sci. 2015, 438, 68–76. 

Li, J.; et al. Gold nanoparticle size and shape influence on osteogenesis of mesenchymal stem cells. Nanoscale 2016, 8, 7992–8007. et al. The softer and more hydrophobic the better: Influence of the side chain of polymethacrylate nanoparticles for cellular uptake. Macromol. Biosci. 2010, 10, 1034–1042.

Mody, N.; et al. Oxidative stress modulates osteoblastic differentiation of vascular and bone cells. Free Radical Biol. Med. 2001, 31, 509–519

Mody, N.; et al. Oxidative stress modulates osteoblastic differentiation of vascular and bone cells. Free Radical Biol. Med. 2001, 31, 509–519

Park, J.; et al. Nanosize and vitality: TiO2 nanotube diameter directs cell fate. Nano Lett. 2007, 7, 1686–1691.

Shrestha, S.; et al. Influence of titanium dioxide nanorods with different surface chemistry on the differentiation of rat bone marrow mesenchymal stem cells. J. Mater. Chem. B 2016, 4, 6955–6966. 

Tekin, H.; et al. Controlling spatial organization of multiple cell types in defined 3d geometries. Adv. Mater. 2012, 24, 5543–5547.

Van, V.T.; et al. mRNA-based gene transfer as a tool for gene and cell therapy. Curr. Opin. Mol. Ther. 2007, 9, 423–431.

Zhang, R.; et al. Silver nanoparticles promote osteogenesis of mesenchymal stem cells and improve bone fracture healing in osteogenesis mechanism mouse model. Nanomed. Nanotechnol. Biol. Med. 2015, 11, 1949–1959. 

Zhang, S.; et al. Physical principles of nanoparticle cellular endocytosis. ACS Nano 2015, 9, 8655–8671.

Zorlutuna, P.; et al. Microfabricated biomaterials for engineering 3d tissues. Adv. Mater. 2012, 24, 1782–1804.

Please explain the trigger principle according to the nanoparticle material (pH, applied magnetic field AC, photo-dynamic interaction, ultrasound, etc.)

pH

Another example of metallic NP-mediated drug delivery is the delivery of the doxorubicin (DOX) through its conjugation with the surface of AuNP via a poly (ethylene glycol) spacer through a pH-sensitive linkage. This delivery model showed efficient delivery and release of DOX, particularly in the multidrug resistant cancer cell line MCF-7/ADR (Chen et al. ,2019).

Chen J, et al. Doxorubicin-conjugated pH-responsive gold nanorods for combined photothermal therapy and chemotherapy of cancer. Bioact Mater. 2018 May 15;3(3):347-354.

The effect of pH in the lysosome is cue for NP degradation within the cell

On the other hand, intracellular degradation of NPs often take place in the lysosomes, where most NPs converge and accumulate following endocytosis pathway. Endolysosomes are characterized by an oxidative potential, a moderate acidity (minimum of pH 4.5), a high proteolytic activity, and redox regulation pathways, including glutathione and MTs metabolism as well as ferritin regulation and storage. This environment favors the dissolution of several metallic NPs, including iron oxides,ZnO CuO, Ag,and Au NPs, and thus intracellular exposure to metallic ions. Lysosomal processing of metal ions has been also described to regulate the metabolism/homeostasis of metals or ensure detoxification through sequestration (Zhang X, et al. Use of acidic nanoparticles to rescue macrophage lysosomal dysfunction in atherosclerosis. Autophagy. 2023,19(3):886-903).

 The case of the recently evidenced Au NP degradation within lysosomes is particularly informative; Since Au NPs are poorly reactive, their dissolution is observed over weeks to months in cells. A longitudinal transcriptomic monitoring during this dissolution process revealed that the cell response to Au NPs was markedly different from that to Au ions at the shortest time point, while both responses converged at the latest time points. This finding suggests that the pace of NP degradation within the cells is a key factor to understand the similarities between the cell responses to metallic NPs and their ionic forms. In addition to these dissolution mechanisms, the commonalities in the responses to ions and NPs can also be explained in light of the recrystallization process of metals inside the cells. As an example, Au ions and NPs have a common intracellular fate as they end up as a single form, called aurosomes, after exposure and internalization by cells. These intralysosomal structures, which appear quickly after exposure to ionic Au, but within days to months after exposure to Au NPs, are composed of self-assembled crystallized Au nanoclusters. Aurosome formation strikingly illustrates that ions and NPs of the same metal can be processed by cells in the same manner, eliciting similar gene expression. Crystallization processes from ionic species have also been described for other metals, such as Pt, Ag, Zn, and Fe.It has been demonstrated that intracellular and extracellular medium continuously transform Ag, Cu, Zn, Fe, and Au NPs into their ionic forms, and, conversely, that cells exposed to metal ions can also biomineralize NPs in situ. Thus, the continuity of intracellular fate can explain that mostly the same genes are involved in the cell response to ionic and nanoparticulate forms of these metals, thus sharing a common metabolism for NPs and ions (for review consult Almeida et al, 2021, Dayem et al., 2018).Interestingly, Rod-shaped AuNP with a size of 70 nm markedly promoted the osteogenic differentiation, while 40 nm rod-shaped AuNPs suppressed osteogenic differentiation (for review  consult Dayem et al., 2018; Hsiao, et al. 2015; Levy, et al. 2011; Strauch, et al 2020;  Balfourier, et al. 2020;  Asano, et al. 2011;  Balfourier, et al. 2020; .Rehman, et al. 2018;  Van de Walle, et al. 2019; Schwartz-Duval, A. S.; et al.. 2020)

Hsiao, I.-L.; et al. Trojan-horse mechanism in the cellular uptake of silver nanoparticles verified by direct intra-and extracellular silver speciation analysis. Environ. Sci. Technol. 2015, 49, 3813−3821.

Levy, M.; et al. Long term in vivo biotransformation of iron oxide nanoparticles. Biomaterials 2011, 32, 3988−3999.

Strauch, B. M.; et al. Impact of Endocytosis and Lysosomal Acidification on the Toxicity of Copper Oxide Nanoand Microsized Particles: Uptake and Gene Expression Related to Oxidative Stress and the DNA Damage Response. Nanomaterials 2020, 10, 679.

Balfourier, A.; et al. Unexpected intracellular biodegradation and recrystallization of gold nanoparticles. Proc. Natl. Acad. Sci. U. S. A. 2020, 117, 103−113.

Asano, T.; et al. Distinct mechanisms of ferritin delivery to lysosomes in iron-depleted and iron-replete cells. Molecular and cellular biology 2011, 31, 2040−2052.

Balfourier, A.; et al. Gold-based therapy: From past to present. Proc. Natl. Acad. Sci. U. S. A. 2020, 117, 22639−22648.

Rehman, F. U et al. Mammalian cells: a unique scaffold for in situ biosynthesis of metallic nanomaterials and biomedical applications. J. Mater. Chem. B 2018, 6, 6501−6514.

Van de Walle, A.; et al. Biosynthesis of magnetic nanoparticles from nano-degradation products revealed in human stem cells. Proc. Natl. Acad. Sci. U. S. A. 2019, 116, 4044−4053

Schwartz-Duval, A. S.; et al. Intratumoral generation of photothermal gold nanoparticles through a vectorized biomineralization of ionic gold. Nat. Commun. 2020, 11, 1−18.

Photodinamic enhances antitumoral effects of copper sulfide (CuS) NPs nanoparticles

AgNP successfully delivered photo-activated miR-148b mimic into the intracellular space without the use of transfection vectors. Upon photo-activation, this construct results in upregulation of osteogenic differentiation-related markers in stem cells (Qureshi, A.T et al., 2013).Likewise, ZnO NP-induced ROS generation was implicated in its anti-cancer effects in human cancer cells (De Berardis., 2010). Interestingly, the selectivity of ZnO NPs in killing human myeloblastic leukemia cells, but not the normal peripheral blood cells has been shown (Premanathan., 2011). This cytotoxic action was ascribed to the potency of ZnO NPs to generate ROS, as well as the induction of ultrasound-mediated lipid peroxidation (Premanathan., 2011).  The unique optical characteristics of copper sulfide (CuS) NPs have also been exploited as an anti-cancer strategy. Irradiation of the CuS NPs with NIR at 808 nm resulted in photothermal-mediated anti-cancer proliferation activity against HeLa cells in a dose-dependent manner (Li 2010). The conjugation of TiO2 NPs with folic acid showed potent anti-cancer effects against HeLa cells (Lai, 2009).

On the other hand, the anti-microbial activity of TiO2 NPs was elevated when combined with gold in an Au/TiO2 nanocomposite, a finding which was attributed to the alteration in the surface charge of TiO2 NPs when conjugated with gold (Arbelao et al., 2007).

Armelao, L.; et al. Photocatalytic and antibacterial activity of TiO2 and au/TiO2 nanosystems. Nanotechnology 2007, 18, 375709

De Berardis, B.; et al. Exposure to ZnO nanoparticles induces oxidative stress and cytotoxicity in human colon carcinoma cells. Toxicol. Appl. Pharmacol. 2010, 246, 116–127.

Lai, T.-Y.; Lee, W.-C. Killing of cancer cell line by photoexcitation of folic acid-modified titanium dioxide nanoparticles. J. Photochem. Photobiol. A 2009, 204, 148–153

Li, Y.; et al. Copper sulfide nanoparticles for photothermal ablation of tumor cells. Nanomedicine 2010, 5, 1161–1171.

Premanathan, M.; et al. Selective toxicity of ZnO nanoparticles toward gram-positive bacteria and cancer cells by apoptosis through lipid peroxidation. Nanomed. Nanotechnol. Biol. Med. 2011, 7, 184–192.

Qureshi, A.T.; et al. mir-148b–nanoparticle conjugates for light mediated osteogenesis of human adipose stromal/stem cells. Biomaterials 2013, 34, 7799–7810

Effect of magnetic fields

SPIO Superparamagnetic iron oxide (SPIO NPs) are a type of iron oxide NPs (IONPs) that possess superparamagnetism property when are subjected to an external magnetic field (LaConte, et al., 2005; Bull, E.; et al., 2014). For example, coating of SPIO with dextran (DEX) and then labeling of stem cells led to a marked induction of myogenic differentiation under pulsed electromagnetic field, which was evidenced through the upregulation of the expression level of the myogenic-specific markers (MyoG and Myh2) (Norizadeh-Abbariki, T.; et al., 2014). SPIO NPs are considered as being a safe nanomaterial for stem cell labelling.Interestingly, chemo-attractants, namely, chemokine receptor type 4 (CXCR4) and epidermal growth factor receptor (EGFR) can recruit DEX-IONP-labeled hMSC to the site of damage (Chung, T.-H et al., 2018).

On the other hand, external stimuli such as tunable magnetic fields can influence the high rate of particle internalization into stem cells (LaConte, L., et al, 2005). In this way, the magnetic field-induced assembly (stripe-like) of magnetic IONPs was exploited for the conversion of primary mouse bone marrow cells into osteoblasts (Sun, J.; et al., 2014). The interface between IONP magnetic assemblies and the cells promote osteogenic differentiation rather than the particles internalization into the human bone marrow stem cells (BM-MSCs). Interestingly, the magnetic field-induced assembly (stripe-like) of magnetic IONPs was exploited for the conversion of primary mouse bone marrow cells into osteoblasts (Sun, J.; et al., 2014). The interface between IONP magnetic assemblies and the stem cells contribute to the induction of osteogenic differentiation rather than the particles internalization into the hBM-MSCs cells (Wang, Q.; et al., 2016); In addition, SPIO NPs induced the proliferation of hMSCs by regulating cell cycle-related proteins under oxidative stress (Huang, D.-M et al., 2019). While IONP-induced gap junction communication between cardiomyoblasts and MSCs protected against myocardial infarction in a rodent model (Han, J et al., 2015).

Bull, E.; et al. A.M. Stem cell tracking using iron oxide nanoparticles. Int. J. Nanomed. 2014, 9, 1641.

Chung, T.-H.; et al. Dextran-coated iron oxide nanoparticle-improved therapeutic effects of human mesenchymal stem cells in a mouse model of parkinson’s disease. Nanoscale 2018, 10, 2998–3007.

Han, J.; et al. Iron oxide nanoparticle-mediated development of cellular gap junction crosstalk to improve mesenchymal stem cells’ therapeutic efficacy for myocardial infarction. ACS Nano 2015, 9, 2805–2819.

Huang, D.-M., et al. The promotion of human mesenchymal stem cell proliferation by superparamagnetic iron oxide nanoparticles. Biomaterials 2009, 30, 3645–3651

LaConte, L.; et al. Magnetic nanoparticle probes. Mater. Today 2005, 8, 32–38.

Levy, I.; et al. Bioactive magnetic near infra-red fluorescent core-shell iron oxide/human serum albumin nanoparticles for controlled release of growth factors for augmentation of human mesenchymal stem cell growth and differentiation. J. Nanobiotechnol. 2015, 13, 34.

Norizadeh-Abbariki, T.; et al. Superparamagnetic nanoparticles direct differentiation of embryonic stem cells into skeletal muscle cells. J. Biomater. Tissue Eng. 2014, 4, 579–585.

 Sun, J.; et al. Magnetic assembly-mediated enhancement of differentiation of mouse bone marrow cells cultured on magnetic colloidal assemblies. Sci. Rep. 2014, 4, 5125.

Wang, Q.; et al. Magnetic iron oxide nanoparticles accelerate osteogenic differentiation of mesenchymal stem cells via modulation of long noncoding rna inzeb2. Nano Res. 2017, 10, 626–642.

Wang, Q.; et al. Response of mapk pathway to iron oxide nanoparticles in vitro treatment promotes osteogenic differentiation of hbmscs. Biomaterials 2016, 86, 11–20.

We included these references in the R1 version. For example, TiO2 nanorods functionalized with various functional groups, such as carboxyl groups (–COOH), poly (ethylene glycol) (–PEG), and amines (–NH2), showed a variation in their uptake by rat bone marrow-derived MSCs (rBM-MSCs) according to data with transmission electron microscopy (TEM) and inductively coupled plasma mass spectrometry (ICP-MS) (Shrestha, et al., 2016).The modification of the particle surface by the addition of functional group can modulate its cellular uptake and consequent toxicity. In fact, the high rate of cellular internalization has been observed in TiO2–NH2 nanorods and the core nanorods, compared with TiO2–COOH and TiO2–PEG nanorods that showed lower uptake. However, the TiO2 core nanorods possessed the most toxic effects via ROS generation, which could be mitigated by the addition of the surface functional groups (Shrestha et al., 2016, Jiang et al., 2016).

Shrestha, S.; et al. Influence of titanium dioxide nanorods with different surface chemistry on the differentiation of rat bone marrow mesenchymal stem cells. J. Mater. Chem. B 2016, 4, 6955–6966

Jiang, P.; et al. Fe3O4/bsa particles induce osteogenic differentiation of mesenchymal stem cells under static magnetic field. Acta Biomater. 2016, 46, 141–150

  1. Please indicate the particle size and shape for each nanoparticle: gold, silica, iron oxide, carbon, zinc oxide, etc.

The size range of Naoparticles (NP) were reviewed by Dolai in ACS Appl. Nano Mater 2021. They indicate that 1–1000 nm is the considered size for nanomaterials and nanoparticles (NPs) (Dolai et al. ¨Nanoparticle size effects in Biomedical applications¨. ACS Appl. Nano Mater. 2021, 4: 6471-6491). Other references on the size of NP were described by Boverhof et al., 2015 and others (Boverhof et al., 2015; Nanotechnologies—Vocab., Part 1 Core terms, 2015; EC, Commission recommendation of 18 October 2011 on the definition of nanomaterial (2011/696/EU), 2011). The plasmodid properties Au/Ag (1-100 nm), fluorescence properties of quantum dots (1-10 nm), magnetic properties of iron oxides (2-50 nm) are described by Dolai et al 2021 and more studies (for review consult Jayanta Dolai, et al. ¨Nanoparticle size effects in Biomedical applications¨. ACS Appl. Nano Mater. 2021, 4: 6471-6491).

Nanoscale unit   major compound    size/shape

-Iron oxide                     Fe3O4      5-10 nm   size-dependent visible emission (fluorescence probe)                                                                                                 

-Iron oxyhydroxide    Iron (III) oxyhydroxide        2-3 nm

-Zn-S-Capped              Cd Se         1-6 nm, size-dependent visible emission (fluorescence  probe). 

- Au                                                  2-100 nm, size-dependent visible  absorption/scattering  (contrast).  

- Au nanorod                                  5-20 nm diameter.

10-100 nm length length dependent absorption/scattering. magnetic resonance (MR)

- Ag              2-100 nm size-dependent magnetic property (MR constrast  agent,  detection probe).

- Dye-doped silica     10-100 nm size-dependent colloidal stability

                                                                                                           (cellular/detection probe).

- Polymeric micelle Au      10-200 nm size-dependent dispersion stability                              (detection probe).

- Polymeric micelle Au     10-200 nm size-dependent dispersion stability (detection probe).

- Au                         10-200 nm size-dependent dispersion stability (detection  probe).

-carbon nanoparticles   1-10 nm diameter, length-dependent solvent extraction

   (cell membrane translocator).

Adapted from Jayanta Dolai, et al. ¨Nanoparticle size effects in Biomedical applications¨. ACS Appl. Nano Mater. 2021, 4: 6471-6491.

Size of each particle

Shape is a physical property that can also influence the uptake of NPs. Xie et al. synthesized different gold nanostars, nanorods, and nanotriangles coated with methylpolyethylene glycol and exposed them to macrophage RAW264.7 cell line. In this study, all AuNPs enter cells through clathrin-mediated uptake. Rods can also be internalized via caveolae/lipid raft-mediated endocytosis; Shape can also have an impact on the increased or decreased uptake of NPs. In fact, sphero-cylinders are more efficiently endocytosed compared to spheres of the same diameter. Even though both shapes have the same kinetic barrier for uptake across a membrane, the sphero-cylinders possess the larger volume. Chithrani et al. found a lower amount of rod-shaped AuNPs within HeLa cells compared to spherical ones. There are many speculations for this outcome, such as differences in the membrane curvature, reduction of the available receptor binding sites, surfactant molecules, which prevent serum proteins from binding onto the NPs surface efficiently, and nonhomogenous protein coating and thus lack of multivalent binding to the receptors (Mauro Sousa de Almeida et al., 2020). SPIONs and silica-coated iron oxide NPs (Fe3O4@SiO2) with negative surface charge and a primary diameter of around 17 to 30 nm were shown to be internalized via CDC42 (CLIC–GEEC pathway) and caveolae in HeLa cells (for review consult Mauro Sousa de Almeida et al., 2020; Dolai et al., 2021).

In general, the size of a substance/particle is considered as one of the most important parameters in endocytosis. Large particles (4500 nm) are known to be internalized only via phagocytosis and/ or macropinocytosis, while the other endocytic mechanisms are limited in terms of cargo size (maximum size of 200–300 nm). Several other uptake mechanisms are also involved in the internalization of NPs. The NP size is not a critical parameter influencing phagocytosis and macropinocytosis mechanisms. In contrast, clathrin- and caveolae-mediated endocytosis seem to be dependent on NPs size. Caveolae based vesicles are usually smaller (60 nm) in comparison with clathrin-based vesicles (120 nm); thus, it is expected that larger NPs preferentially are taken up by the cells via clathrin. Ho et al. revealed that 20 and 40 nm PS NPs are more dependent on caveolae-mediated endocytosis than 100 nm PS NPs, as seen in HUVEC cells. Similar observations were reported in hepatocytes (HepG2) where the uptake of 20 nm AuNPs showed a higher dependence of caveolae in comparison to 40 nm AuNPs. In addition, studies on BLMVEC revealed that a single caveolae vesicle was able to engulf up to three 20 nm or two 40 nm albumin-coated polymeric NPs.An important size-related parameter that can affect NP internalization is the aggregation state, once NPs in an aggregated form or as individual particles interact differently with cells. The NP aggregation contributes to an overall increase in NP size and may affect their uptake and intracellular distribution. For example, Halamoda-Kenzaoui et al. showed that well-dispersed SiO2 NPs were internalized principally through caveolae-mediated endocytosis, but an increase in the NP agglomeration state shifted to a combination of endocytosis pathways with a predominant role of macropinocytosis. Size is not only related to the route of internalization but also affects the uptake rate. In fact, the internalization of SiO2 NPs in lung epithelial cells becomes slower with increasing particle size. The same observation was reported by Rejman et al. where the uptake of fluorescent carboxylate nano-/microspheres (50, 100, 200, 500, and 1000 nm), by melanoma cells, revealed to be size-dependent. A decreased internalization was related to increased microsphere size. In contrast, an increase in the uptake of larger NPs was observed in a different study where AuNPs (13 and 45 nm) were exposed to human dermal fibroblasts. These authors demonstrated that 45 nm AuNPs enter the cell more efficiently than 13 nm AuNPs. Yet Li et al. demonstrated the interplay between different-sized SiO2 NPs (50, 100 and 150 nm) on their uptake in HeLa cells in a co-exposure scenario. When NPs were administered simultaneously to the cells, a competition between different-sized NPs in their cellular uptake was observed. Interestingly, the bigger NPs stimulated the uptake of smaller ones and vice versa.  Rafieepour et al. investigated the in vitro toxicological effects of single and combined exposure of magnetite (Fe3O4) NPs and polymorphous silica NPs on human epithelial cell line A549.367 The cells were exposed to four different NP concentrations (10, 50, 100, and 250 mg/mL) of both NP types simultaneously for 24 h and 72 h. The data obtained in this study showed that increasing the concentration and exposure time to Fe3O4 and SiO2 NPs individually increased toxic effects. In contrast, the effect of combined exposure to Fe3O4 and SiO2 NPs led to antagonistic interactions (for review consult Mauro Sousa de Almeida et al., 2021; Dolai Jayanta et al., 2021; Alice Balfourier et al., 2022; Abdal Dayem, A et al.,120; Daymen et al., 2017; Alice Balfourier  et al., 2022)

Abdal Dayem, A.; et al. The role of reactive oxygen species (ros) in the biological activities of metallic nanoparticles. Int. J. Mol. Sci. 2017, 18, 120.

Alice Balfourier , et al. Importance of Metal Biotransformation in Cell Response to Metallic Nanoparticles: A Transcriptomic Meta-analysis Study. ACS Nanosci Au 2022 Sep 30;3(1):46-57.;

Almeida, et al. Understanding nanoparticle endocytosis to improve targeting strategies in nanomedicine. Chem. Soc. Rev., 2021, 50, 5397

Dolai Jayanta, et al. ¨Nanoparticle size effects in Biomedical applications¨. ACS Appl. Nano Mater. 2021, 4: 6471-6491.

  1. The challenges and perspectives should be included at the end of this review.

We have added these perspectives at the end of the review (conclusion and future perspectives).

Thanks again for your comments.

Round 2

Reviewer 1 Report

Dear Editor of Micromachines

Corrections have been made and the article is ready for publishing.

Best Regards

Author Response

Thanks to you as well as to all reviewers for their comments again, which improves this manuscript.

This time, we have marked small corrections in blue color as well as the table-2 has been improved following reviewer-3 advice (marked in blue color).

I undestand everything has been solved at this time

Thanks a lot.

Reviewer 3 Report

Dear Authors,

The manuscript Nanoparticles and mesemchymal stem cell (MSC) teraphy for cancer treatment: focus on nanocarriers and si-RNA chemokine CXCR4 blockade as strategies for tumor erradication in vitro an in vivo has been improved. This can be consider by the Micromachines MDPI Journal as Accepted after minor revision.

Please consider:

(1) SPIONs abbreviation must be used to replace SPIO NP, IONPs, SPIO NPs notations for lines 381 to 414. 

(2) Please add information about the NPs in table 2 (Shape: nanospheres, nanorods, nanocubes....etc. Configuration: core-shell, capping, thin film, etc.)

Best Regards

Dear Editor,

Quality of English is good, the style corrections can be considered in the final edition.

Author Response

Dear reviewer.

Thanks your comments, which help us to improve the manuscript content.

  • SPIONs abbreviation must be used to replace SPIO NP, IONPs, SPIO NPs notations for lines 381 to 414. 

Done it¡. I have replaced SPIO NP, IONPs, SPIO NPs notations by SPIONs abbreviation

  • Please add information about the NPs in table 2 (Shape: nanospheres, nanorods, nanocubes....etc. Configuration: core-shell, capping, thin film, etc.)

We have included this inforamtion in the table-2. Thanks again for your comments.
